# Growth rate-driven modelling suggests that phenotypic adaptation drives drug resistance in BRAFV600E-mutant melanoma

Sara Hamis [1] ✉, Alexander P. Browning [2], Adrianne L. Jenner[3], Chiara Villa[4,5], Philip K. Maini [2] &
Tyler Cassidy[6]

Phenotypic adaptation, the ability of cells to change phenotype in response to external pressures, has been identified as a driver of drug resistance in cancer. To quantify phenotypic adaptation in BRAFV600E-mutant melanoma, we develop a theoretical model informed by growth-rate data of WM239A-BRAFV600E cells challenged with the BRAF-inhibitor encorafenib. We use an individual-based model (IBM) in which each cell is described by one of multiple discrete and plastic phenotype states that are directly linked to drug-dependent net growth rates and, by extension, drug resistance. To describe how cells transition between phenotype states, we explore a gamut of candidate models common in the mathematical biology literature. Comparing these on their ability to reproduce in vitro growth curves, data-matched simulations suggest that phenotypic adaptation is directed towards states of high net growth rates, enabling the evasion of drug-effects. The model subsequently provides an explanation for when and why intermittent treatments outperform continuous treatments in the studied system, and demonstrates the benefits of not only targeting, but also leveraging, phenotypic adaptation in treatment protocols. Building on the IBM, we present a flexible mathematical methodology based on ordinary differential equations to compare responses to continuous and intermittent treatments through long-term effective net growth rates.

Over 50% of melanoma cases present with BRAFV600 mutations that result in constitutive activation of the Mitogen Activated Protein Kinase (MAPK)[1,2]. Consequently, targeted small molecule therapies that selectively inhibit mutant BRAF signalling have been developed over the last decades and are now part of the standard of care for BRAF-mutant melanoma. While these targeted therapies have led to a remarkable increase in progression free survival, treatment resistance inevitably develops and limits long-term survival[3]. This drug resistance in BRAF-mutant melanoma, and other solid tumours, is increasingly understood as a result of intratumoural heterogeneity, which has long been viewed through the lens of inter-clonal differences[4–6]. However, a growing body of work implicates cell plasticity, i.e., phenotypic adaptation without associated genetic mutations, as a key component of therapy resistance in melanoma[1,7,8].

Advances in transcriptomics and multi-omics have led to the identification of reversible and adaptive phenotype changes that confer treatment resistance and are mediated by a number of complex physiological factors on the single-cell level[9–11]. As systems-level experiments of the complex interactions driving these adaptations are currently intractable, computational models have become increasingly important to our understanding of how therapeutic selection pressure shapes the epigenetic evolution of malignant tumours[12–14]. Such evolution is driven by epigenetic changes in cancer cells, i.e., changes in gene expression that do not involve changes in the cells' DNA sequence[15].

Phosphorylated, i.e., activated, ERK is often considered to be the main output of interest in pathway-level studies of BRAF-mutated melanoma. This follows from the fact that activated ERK phosphorylates multiple

[1]Department of Information Technology, Uppsala University, Uppsala, Sweden. [2]Mathematical Institute, University of Oxford, Oxford, UK. [3]School of Mathematical Sciences, Queensland University of Technology, Brisbane, QLD, Australia. [4]Sorbonne Université, CNRS, Université de Paris, Inria, Laboratoire Jacques-Louis Lions UMR, Paris, France. [5]Centre Inria de Saclay, Université Paris-Saclay, Inria, Palaiseau, France. [6]School of Mathematics, University of Leeds, Leeds, UK.
✉e-mail: sara.hamis@it.uu.se

proliferation-promoting substrates in the nucleus and cytoplasm, which can drive uncontrolled cell growth and division[16]. Consequently, many existing computational approaches informed by extensive perturbation experiments leverage precise mechanistic representations of the MAPK pathway to describe signalling dynamics culminating in phosphorylated ERK[13,17]. With this ERK-endpoint, such models are not explicitly linked with the cellular response. Further, due to the large number of chemical reactions being modelled, these models are computationally expensive to train and simulate. Here, we present a novel approach to understanding cell plasticity that is computationally inexpensive and directly bridges cellular phenotype with net growth rates. Specifically, we develop a computational model that is directly informed by growth rate analysis of BRAFV600E-WM239A melanoma cells exposed to the BRAF-inhibitor encorafenib (LGX818) in a recent study by Kavran et al.[9].

In our model, each individual cell is described by a plastic phenotype state that can take one of $S$ discrete values, where $S$ is an arbitrary integer larger than 0. Further, each phenotype state is directly related to drug dose-dependent growth rates. Modelled phenotype states are thus mathematical abstractions that link cells to drug resistance. To investigate phenotypic adaptation in response to drug exposure and removal, we study four candidate cell-level adaptation strategies that are commonly used in the mathematical modelling literature. Subsequent comparisons of model-generated cell populations and in vitro cell count data reveal that phenotypic adaptation is directed towards phenotype states with high net growth rates in the regarded cell system. As such, our individual-based model demonstrates the emergence of drug resistance at the population-level as a result of directed phenotypic adaptation on the individual cell-level. Further, our modelling captures the experimental observation that intermittent treatment strategies can outperform their continuous counterparts in vitro, despite halved cumulative drug doses[9]. Based on multiple-dose growth rate data from untreated and treated cells, our framework thus provides a mechanistic explanation (on the phenotype classification level) for when and why intermittent treatments may be more effective than continuous treatments. Building on these results, we perform further simulation experiments and mathematical analysis to quantitatively assess the potential of not only targeting, but also leveraging, cell plasticity in melanoma towards improved clinical outcomes.

## Results
### Empirical growth rates are mapped to phenotype states and give rise to a theoretical model
Our theoretical model formulation emerges from data analysis of drug dose-dependent net growth rates of BRAFV600E-mutant melanoma cells exposed to the BRAF-inhibitor encorafenib. The growth rates are shown in Fig. 1a and are extracted from a recent experimental study by Kavran et al.[9] in which cell count numbers of WM239A-BRAFV600E melanoma cells were measured at 0 and 72 h in response to drug challenge following 0–4 weeks of 500 nM encorafenib incubation. Cell populations that have been continuously incubated with drug for 1–4 weeks present dose-dependent net growth rates that are not significantly different from each other. We therefore assume that cells reach a drug-adapted state within 1 week of encorafenib incubation and stratify the observed growth rates as belonging to one of two extreme phenotypes: drug-naive cells and drug-adapted cells, i.e, cells that have been incubated in 500 nM encorafenib for 0 and 1–4 weeks, respectively (Fig. 1b). Further, upon drug removal, drug-adapted cells in the regarded in vitro experiment resumed growth rates similar to those of drug-naive cells[9] (results not shown) hence we allow for bi-directional phenotype updates in the model.

Building on our growth rate stratification, we formulate a discrete adaptive phenotype model in which each individual cell in a cell population is described by a discrete and plastic phenotype state that can be updated in response to phenotypic instability or external pressures, here encorafenib presence or absence. We use the variable $x_i$ to describe the phenotype state of cell $i$, where a cell's phenotype state corresponds to its sensitivity to the BRAF-inhibitor such that, arbitrarily, cells in state $x = 0$ are the most drug-

sensitive and cells in state $x = 1$ are the most drug-resistant. In our model, growth rates for these phenotypes correspond to those for drug-naive and drug-adapted cells in Fig. 1b, respectively. We further assume that the cells can exist in $n$ intermittent phenotype states between the extrema $x = 0$ and $x = 1$, where each state is associated with drug dose-dependent net growth rates that are estimated through linear interpolation of growth rate data and are mapped onto a fitness matrix (Fig. 1c, d). The multi-step phenotypic adaptation model is based on the observation that protein expression for L1CAM, a marker for drug resistance in melanoma, changes gradually in response to external pressures in the regarded experiments[9]. In the main body of this article, we arbitrarily use $n = 9$, but key results are shown in the Supplementary Material (SM1) for other choices of $n$, including $n = 0$ which results in a single-step adaptation model.

In our discrete adaptive phenotype model, positive values in the fitness matrix correspond to daily probabilities that cells divide, whereas negative values correspond to daily probabilities that cells die. Note that the fitness matrix approximates net growth rates from the three-day drug-challenge assay for drug-naive cells ($x = 0$), cells pre-incubated at 500 nM encorafenib ($x = 1$), and intermediate phenotype states $x \in (0, 1)$. In Fig. 1d, the yellow line separates doses above and below the pre-incubation dose of 500 nM, highlighting that drug-resistant phenotypes may continue to increase their resistance beyond the three-day assay period at doses above 500 nM. Likewise, cells exposed to doses below 500 nM encorafenib may continue to adapt their net growth rates after the three first days. The absence of longer-term growth rate measurements is a limitation of our study, and in the Discussion we outline how future experiments could be designed to address this and yield more informative fitness matrices.

As is discussed in the next sections, our presented growth rate-to-phenotype state mapping enables growth rate-driven phenotype characterisation of cells, which we use to elucidate the directionality of phenotypic adaptation, predict treatment responses, and investigate the possibility of both targeting and leveraging cell plasticity in treatments.

### Phenotypic adaptation in BRAFV600E-mutant melanoma is directed towards high fitness
Recently, it was observed that four weeks of intermittent (one week on/one week off) treatments with 500 nM encorafenib are more effective at suppressing WM239A-BRAFV600E cell counts than four weeks of continuous treatments with the same dose in vitro[9]. In this work, we set out to elucidate the cell-level behaviour related to phenotypic adaptation that gives rise to this macroscopic, population-level result. To investigate the directionality of phenotypic adaptation in the regarded melanoma cells in response to encorafenib exposure and removal, we present four candidate phenotype update strategies for describing cell-level dynamics. These are illustrated in Fig. 2a and are mathematically defined in the Methods section.

With the *no update* strategy, cells never update their phenotype states. With the *unbiased* strategy, cells propose to update their phenotype states to adjacent states with equal probability, unless at a boundary, and subsequently move to the proposed state. With the *semi-biased* strategy, cells propose to update their phenotype states to adjacent states with equal probability, but only accept the move if it results in a higher fitness, i.e., net growth rate. This corresponds to the cells having a noisy measurement of the fitness gradient. Finally, with the *biased* strategy, cells update their phenotype states to states of higher fitness whenever possible. As such, phenotypic adaptation is non-directed in the first two (no update and unbiased) strategies, but directed in the last two (semi-biased and biased) strategies. Biologically, the unbiased PU strategy could correspond to phenotypic instability, while the biased PU strategy corresponds to stress-induced phenotypic adaptation, in accordance with work by Chisholm et al.[18]. Ultimately, the mechanisms responsible for the development of resistance are not yet fully known, and we therefore also explore the more general semi-biased PU strategy than encompasses both phenotypic instability and adaptation. This gamut of phenotype progression is highly studied in segments of the mathematical literature[19] but, to the best of our knowledge, has not been applied in the interpretation of experimental data to select between

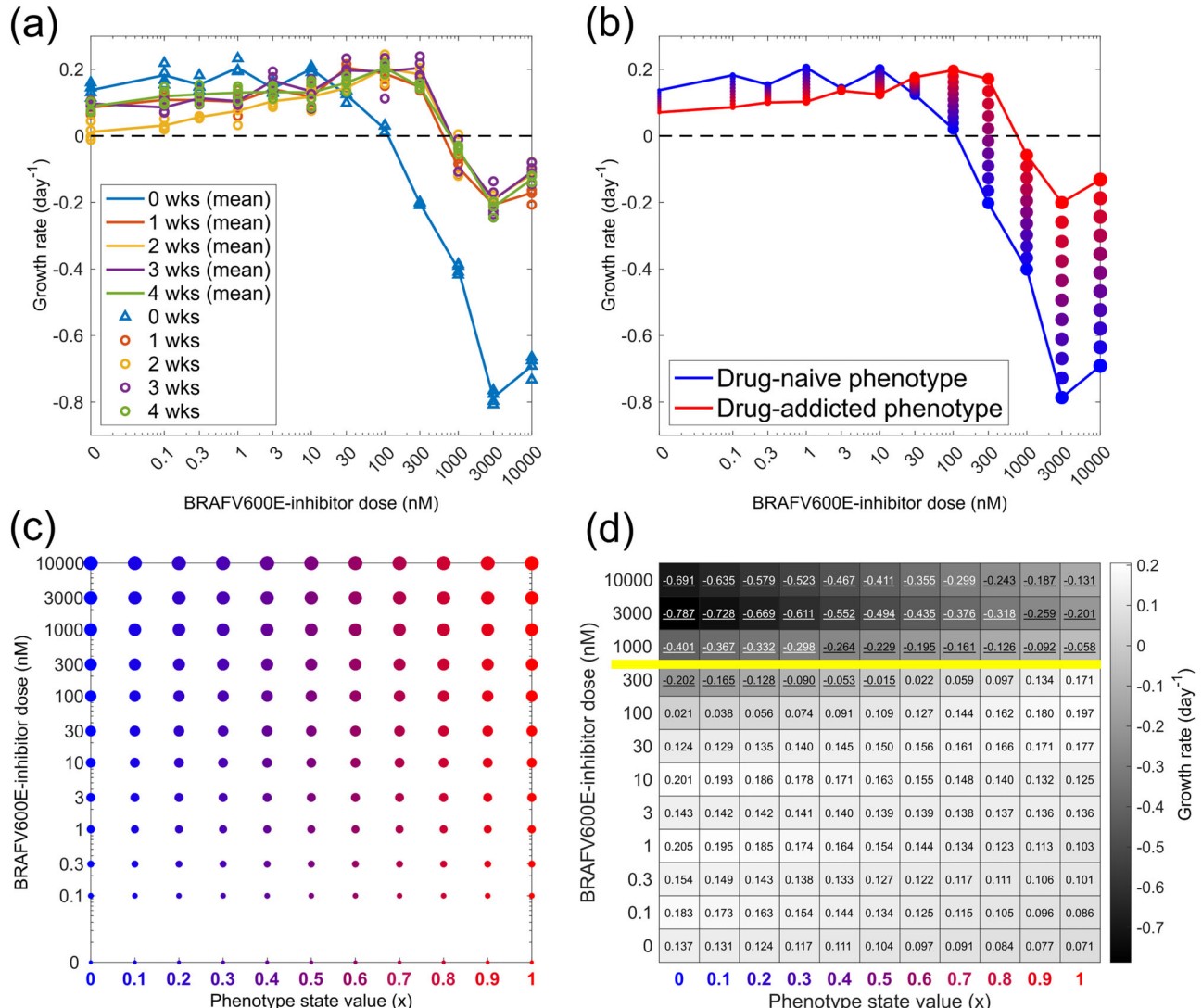

**Fig. 1 | Empirical growth rates are mapped to phenotype states and give rise to a theoretical model. a** Growth rates for BRAFV600E-mutant melanoma cells from the WM239A cell-line are shown in response to various doses of the BRAF-inhibitor encorafenib, as measured over 72 h. The rates are extracted from quadruplicate cell count assays by Kavran et al.[9] Prior to drug challenge, the cells are incubated in 500 nM encorafenib for 0,1, 2, 3, or 4 weeks (wks), with four replicates per incubation time. **b** Mean growth rates for drug-naive phenotypes (0 wks in **a**), and drug-adapted phenotypes (1–4 wks in a) are plotted over drug doses. Between the two extreme phenotype states, $n$ intermediate states are introduced (here, $n = 9$). Marker colours and sizes correspond to phenotype state values and drug doses, respectively. **c** Net

growth rates in (**b**) are mapped onto a discrete phenotype-dose space, as is indicated by marker colours and sizes. **d** Points in phenotype-dose space in (**c**) are mapped onto a fitness matrix (FM). The values in the left-most and right-most columns correspond to three-day growth rates in drug-naive ($x = 0$) and drug-adapted ($x = 1$) phenotype states, respectively resulting after encorafenib pre-incubation at 0 and 500 nM (**b**). Other values in the fitness matrix are obtained via linear interpolation between FM($x = 0, d_j$) and FM($x = 1, d_j$) for each of the $j$ doses. The yellow line separates doses above and below the pre-incubation dose of 500 nM. Negative net growth rates are underlined.

candidate phenotype update models. Note that, while the no update strategy suggests that the presence of drug-resistant cells that exist post treatment arise purely from pre-exisiting phenotypic heterogeneity, the other three strategies allow drug-resistant cells to arise through phenotypic plasticity during treatment.

The four update strategies are implemented through a simple computational algorithm (Fig. 2b) and produce significantly different dynamic phenotype density distributions (see the Methods section for a visual demonstration of this). In the code implementation, the same update rule applies both on and off treatments, but the directionality of the updates in the semi-biased and biased phenotype update strategies is drug dose dependent.

In Fig. 2c, simulated cell counts are shown over time in response to continuous and intermittent treatments with doses 300 and 500 nM. In this figure, $\rho$, which denotes the probability that a cell changes its phenotype state

at a programmed update time, is fixed at $\rho = 1$ and simulated cells have a chance to update their phenotype state $\eta_{off}$ and $\eta_{on}$ times per day in drug absence and presence, respectively. In the 300 nM treatment simulations, $\eta_{on} = \eta_{off} = 2$, ensuring that cells can traverse phenotype space within a week (following observations in Fig. 1a) and dose-dependent net growth rates from the fitness matrix have been used. Comparing simulated cell counts for the two 300 nM scheduling options, our model predicts a distinct benefit in using 300 nM encorafenib intermittent over continuous treatments for the directed phenotype update strategies, but not for the non-directed strategies. For the 500 nM treatment responses in Fig. 2, phenotype-dependent net growth rates and $\eta_{on} = \eta_{off}$ are simultaneously fit to data through a least-squares parameter estimation for each update strategy. In the fitting, net growth rates are bounded by those for the adjacent drug doses for which data is available i.e., 300 nM and 1000 nM (details about the parameter fitting are available in the Methods section). The simulation results with

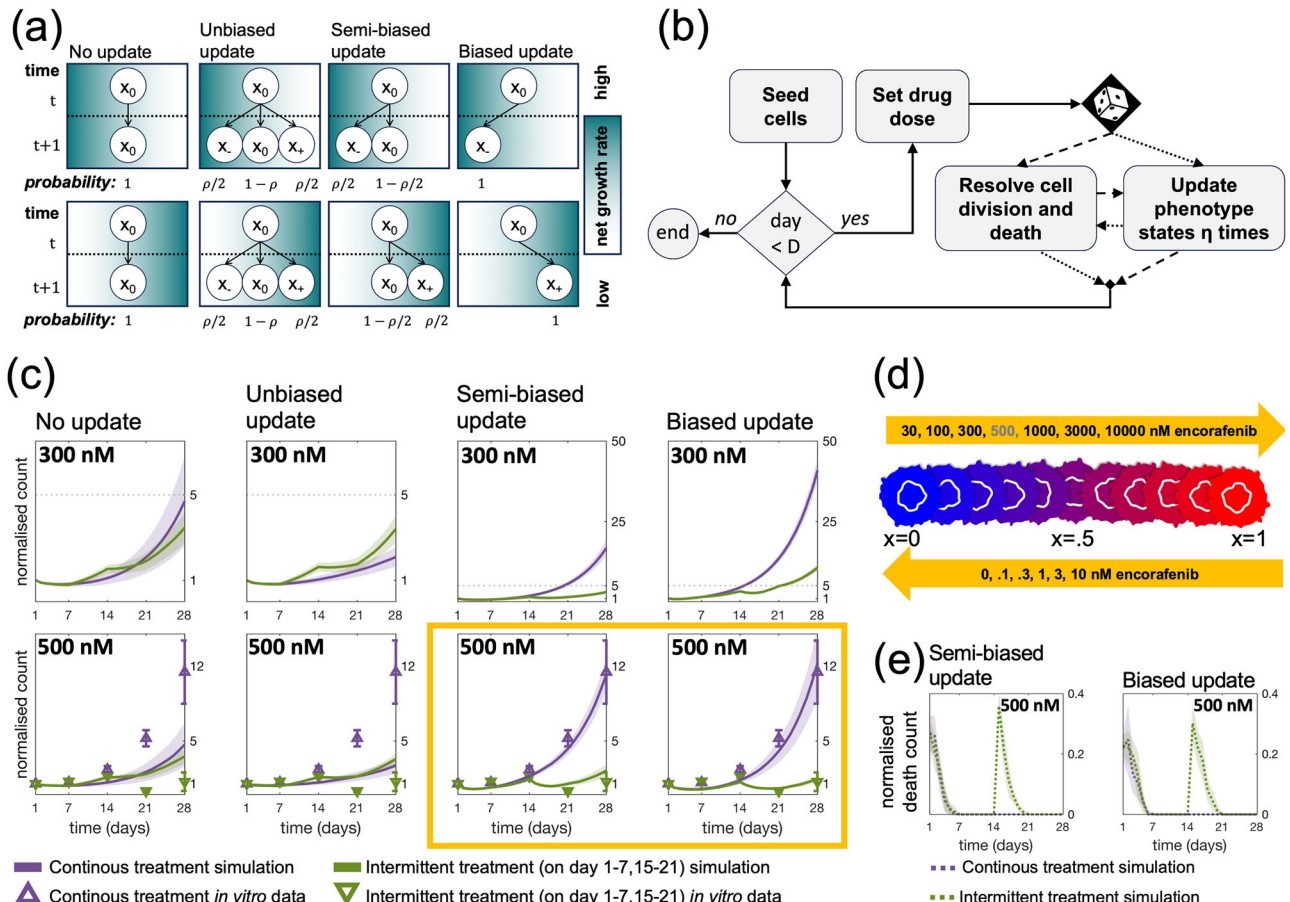

**Fig. 2 | Phenotypic adaptation in BRAFV600E-mutant melanoma cells is directed towards states of high fitness. a** We propose four phenotype update strategies. The figure shows probabilities that a cell in state $x_0$ at time $t$ updates to states $x_-$, $x_0$ and $x_+$ at time $t + 1$ whenever possible, i.e., ensuring that the phenotype states lie between the extrema $x = 0$ and $x = 1$. Net growth rates increase with decreasing $x$-values in the top row, and with increasing $x$-values in the bottom row. The directionality of the net growth rates, i.e., whether the top or bottom row applies, depends on the applied drug dose and is inferred from growth rate data. **b** A simple algorithm is used to implement our discrete adaptive phenotype model with the update strategies in (**a**). Each simulated day, cells update their phenotype states $\eta$ times either before (with probability 0.5) or after (with probability 0.5) the resolution of cell division and death. Probabilistic path branching is implemented to minimise numerical bias arising from the ordering of birth-death and phenotype-update events. **c** The plots show normalised cell counts over time in response to 300 and 500 nM continuous and intermittent treatments. Simulations start with 100 cells at day 1, with update- rule specific initial phenotype distributions matching day 1 in Fig. 4. Simulated mean counts (solid lines) and standard deviations (shaded bands) are shown for 100 simulation runs. Mean (triangles) and standard errors (bars) for 6 experimental in vitro replicates are shown. The yellow box highlights that only the directed strategies (semi-biased and biased updates) are able to capture in vitro cell count dynamics. **d** The highlighted result in (**c**), together with (**a**), suggest that phenotypic adaptation is directed towards states of high fitness, where this direction depends on the applied drug dose. The schematic shows how cells can adapt from more being drug-sensitive phenotypes (blue) to more drug-resistant phenotypes (red). Drug doses included in the top and bottom arrows respectively induce phenotypic adaptation in the direction of increasing and decreasing $x$-values. **e** The plots show the number of death events normalised over the number of cells over time for the simulation experiments in (**c**) with directed update strategies in response to 500 nM encorafenib. Simulated mean counts (dashed lines) and standard deviations (shaded bands) are shown for 100 simulation runs.

500 nM demonstrate that only the directed strategies are able to match in vitro data[9] and reproduce cell population dynamics such that 500 nM encorafenib intermittent treatments outperform continuous treatments, in terms of suppressing cell counts (Fig. 2d).

For all update models, with the exception of the biased update model, cell counts in response to continuous and intermittent treatments are similar in response to 300 nM and 500 nM encorafenib. This result is further discussed and visualised in the Supplementary Material (SM1). The Supplementary Material (SM1) also includes simulated cell counts in response treatments for different choices of phenotype granularity, i.e., $n$, showing that if we consider the cell population to be phenotypically non-binary, then the directed strategies better explain the observed data than do the non-directed strategies.

Taken together, our results suggest that phenotypic adaptation in the regarded WM239A-BRAFV600E cells is directed towards phenotype states of high fitness through semi-biased or biased update strategies (Fig. 2d). With

the used model and available data, we can not distinguish which of the two directed strategies best describe the model system. However, the conclusion that phenotypic adaptation is directed provides a growth rate data-driven explanation for why intermittent treatments here outperform continuous treatments at 500 nM encorafenib over four weeks: the intermittent treatment allows the cells to traverse phenotype space to drug-sensitive states during drug-off periods, and back to drug-resistant states during drug-on periods. Meanwhile, the continuous treatment traps the cells in highly drug-resistant states. Thus, for the cells, the dynamic adaptation induced by intermittent treatment confers a long-term proliferation disadvantage compared to continuous treatment in the regarded BRAF-mutated melanoma system, with cells pre-incubated at 500 nM encorafenib. This follows from the empirical observation that net growth rates are higher for drug-naive cells when the drug is off, but higher for drug-adapted cells when the drug is on.

Moreover, with directed phenotypic adaptation, our model predicts that death events spike once treatments resume after a drug holiday, when

the cells have acquired drug-sensitive phenotype states, in the intermittent treatments (Fig. 2e). This simulation result is in qualitative agreement with Kavran et al.[9] experimental observation that the percentage of propidium iodide (PI) positive cells peak during drug re-challenge, where PI is used as a cell death marker, giving further credence to our model formulation and our simplifying model assumption that positive and negative net growth rates, respectively, can be approximated to yield birth and death events on the cell-level.

## Computational and mathematical modelling demonstrates the benefits of targeting and leveraging cell plasticity in treatments

Our results raise the question: Given directed phenotypic adaptation, when do intermittent treatments outperform their continuous counterparts in vitro? As a step to address this question computationally and mathematically for the studied melanoma system, we perform a series of simulation experiments in which output dynamics are quantified in response to variations of two model-inputs: the number of times per day that a cell can update its phenotype state ($\eta$), and the duration of treatment on/off intervals ($t_{on}/t_{off}$). Varying these parameters allows us to investigate the impact that they have on simulated treatment responses. To enable biologically interpretable and mathematically tractable comparisons of continuous and intermittent treatments, we also introduce the effective net growth rate $\lambda_{eff}$ as the average (per cell and day) net growth rate that cells experience over the course of a simulated treatment. Using the biased update strategy, we derive analytical approximations, formulated through an ordinary differential equation (ODE) model, of effective net growth rates $\lambda_{eff}$ obtained in the limit of large total treatment times (see the Methods section for details). By comparing the $\lambda_{eff}$ values resulting from continuous and intermittent treatments, we can evaluate which treatment is more effective at keeping $\lambda_{eff}$ low, and thus impeding cell population growth. While we, in Fig. 3, focus on intermittent treatments with 1:1 scheduling such that $t_{on}=t_{off} = T$, the developed ODE formulation and methodology can be adapted to study systems with arbitrary schedules and net growth rates.

Using both the simulation and the ODE approach with instantaneous drug on/off switches, we here identify ($\eta$, $T$) pairings for which 1:1 intermittent treatments yield lower effective growth rates than their continuous counterparts. For these ($\eta$, $T$) pairings only, simulated intermittent treatments thus outperform continuous ones long-term, despite resulting in approximately half of the accumulated applied drug. Demonstrating this in Fig. 3b, e, ($\eta$, $T$) pairings for which intermittent treatments outperform continuous ones are highlighted in yellow-coloured heatmap regions, whereas such pairings for which the opposite is true are highlighted in black-coloured regions. In the heatmaps, the two regions are separated by overlaid ODE-derived curves that show when the two treatment strategies yield the same effective growth rate. Note that in the 300 nM heatmaps, all visualised ($\eta$, $T$) pairings outperform the continuous treatment strategy, hence no separating curve is shown. Note further that throughout Fig. 3, we have used phenotype-dependent net growth rates for 500 nM that are obtained through log-linear interpolation between net growth rates for 300 and 1000 nM in the fitness matrix (Fig. 1a). This interpolation is used due to three-day growth rates in 500 nM being missing in the data, and likely does not capture the full drug-adaptation of the cells which contributes to the visual differences between the 300 and 500 nM heatmaps in Fig. 3.

From Fig. 3a–c, we see that when all cells start in the most drug-sensitive phenotype state, the effective net growth rates increase with the number of cell updates per day for the regarded doses and simulation inputs. In other words, intermittent treatments become more effective when $\eta$ is decreased in these cases. As such, our simulations highlight the therapeutic benefit of targeting phenotypic adaptation by slowing it down. Next, motivated by clinical studies in which drug re-challenge shows anti-tumour activity in patients who have previously been treated with BRAF-inhibitors[3,20,21], we investigate the impact of drug-holidays on treatment responses. We do this by performing a simulation experiment in which all cells start in the most drug-resistant phenotype state (Fig. 3d–f), as would

follow from a period of continuous treatment. In an effort to push drug-resistant cells to drug-sensitive states in such cases, we can schedule a drug holiday and delay the treatment restart time. The benefits of delaying intermittent treatments are shown in Fig. 3e, where the most effective treatment schedules are those that allow time for cells to traverse phenotype space to reach drug-susceptible states. Our simulation results thereby highlight the benefit of leveraging phenotypic adaptation in treatments.

In Fig. 3e, f, it is clear that our model produces non-monotonic relationships between effective net growth rates and the number of daily cell updates. For instance, we see that if the treatment interval length is $T = 1$ day, and the first drug-on period starts after day 1 or 10, then cells with $\eta = 1$ are more treatment-sensitive than cells with $\eta = 4$. However, the opposite is true when the drug-on period starts after day 4. These quantitative specifics follow from the chosen data and model; in particular, whether cells have (or do not have) time to traverse phenotype space in a way that makes them sensitive to sudden changes in drug exposure and removal strongly influences which $T$ and treatment-delay values best suppresses cell proliferation from a human perspective, or equivalently, which $\eta$ value best promotes survival for the cells. Although the quantitative results presented in Fig. 3 are problem-specific, the qualitative demonstration that treatment scheduling together with phenotypic adaptation shapes optimal strategies has broader relevance beyond this study and motivates treatment designs that includes monitoring if and how phenotypic expression changes during treatment.

## Discussion

Cell plasticity is not listed as one of the original *Hallmarks of Cancer* which were famously collated in 2000[22]. However, based on the last two decades' research advances, an updated version of the work identifies cancer cells' ability to unlock phenotypic adaptation as an emerging hallmark, intended to stimulate debate amongst researchers and inspire investigations that will improve our understanding of cancer[8]. The work presented in this article directly contributes to this understanding by evidencing that phenotypic adaptation in BRAFV600E-mutant melanoma is directed to states of high fitness, and demonstrating that this adaptation enables evasion of drug effects.

Our model emerges from growth rate data analysis of BRAFV600E-mutant melanoma cells exposed to the BRAF-inhibitor encorafenib. Our conclusion that phenotypic adaptation in the regarded cells is directed towards states of high fitness is obtained through a model-selection procedure, in which a gamut of phenotype update strategies on the cell-level are evaluated against dynamic cell count data in response to intermittent and continuous encorafenib treatments. One notable consequence of our results is that dynamic phenotypic adaptation alone suffices to explain why intermittent treatments can outperform continuous treatments. This explanation offers an alternative to variations of the more common inter-clonal competition-models in which two or more distinct subpopulations of cells, such as a drug-resistant and drug-sensitive clones, interact with each other and compete for resources[23]. This ecological perspective has served as the theoretical underpinning of *adaptive therapy*, where treatment is designed to maintain a drug-sensitive cancer cell population that competitively excludes drug-resistant clones[24,25]. This existing framework has historically relied on an assumed cost-of-resistance, where drug-resistant clones are less fit than drug-sensitive clones in the absence of treatment. However, recent theoretical work has demonstrated that this assumption is not necessary for adaptive therapies to outperform conventional maximum-tolerated dose strategies[26,27]. Here, in contrast with much of the adaptive therapy literature[28,29], we use a simple model to capture improved treatment responses to intermittent therapy without explicitly modelling competition between clones.

The model developed in this study was intentionally kept simple to enable direct mapping between phenotype states and growth rates. While this simplicity facilitates interpretability and has yielded informative results, it also highlights directions for more sophisticated model-experiment integration that could provide deeper quantitative insight into phenotypic adaptation. Future experimental designs could explicitly separate

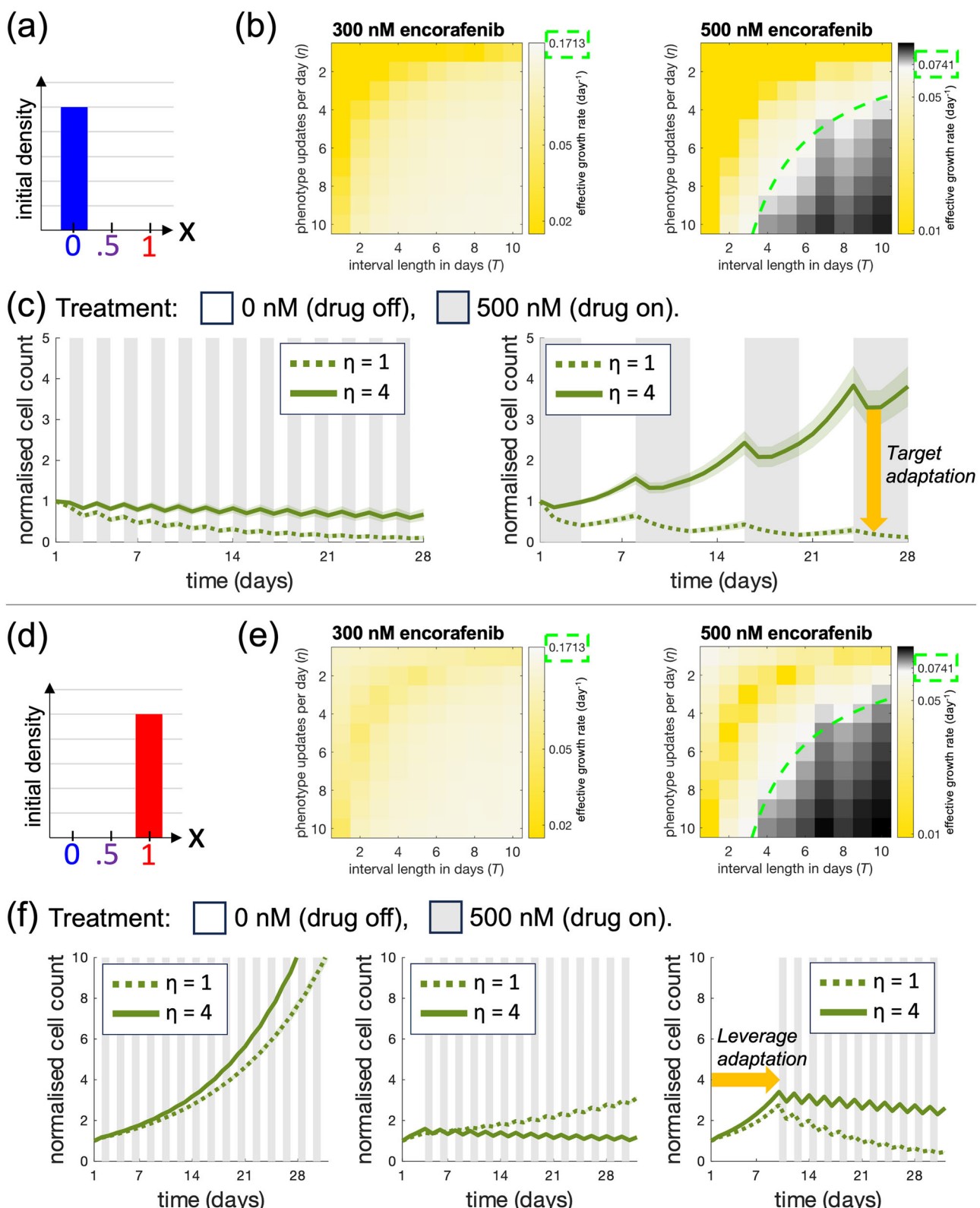

measurements of cell division and death events, enabling models to incorporate growth and death rates independently rather than relying on net growth rates alone. Further, our current approach of linearly interpolating between drug-naive and drug-adapted growth rates to assign fitness to intermediate phenotype states could be refined by systematic drug-challenge assays. Specifically, measuring cell counts at multiple time points after pre-assay incubation across a range of drug doses would allow

the mapping of growth rates in the phenotype-dose parameter space. It would also be informative to test whether, and under which conditions, drug-adapted cells can fully regain the growth rates of drug-naive cells. Such data would permit estimation of dose-dependent adaptation velocities and directly inform ongoing debates on whether cell-level drug resistance is reversible in different cancer types[30]. Incorporating sequencing alongside these assays would further aid in evaluating underlying molecular changes,

**Fig. 3 | Simulations demonstrate the benefits of targeting and leveraging cell plasticity in treatments. a** We consider an initial condition in which all cells are in the most drug-sensitive phenotype state ($x = 0$) in (**a–c**). **b** Effective growth rates in response to 8 week intermittent treatments with 300 nM (left) and 500 nM (right) encorafenib are obtained through simulations with the biased phenotype update strategy. Based on mean values from 100 simulations, heatmap bins show effective growth rates in response to different phenotype updates per day ($\eta$), and on/off treatment intervals ($T$). Effective growth rates for 300 nM (left) and 500 nM (right) continuous treatments are highlighted in the colour legends. In the yellow-coloured bins, intermittent treatments produce long-term effective growth rates lower than those from continuous treatments, and in the black-coloured bins the opposite is true. In the white bins, intermittent and continuous treatments yield approximately equal effective growth rates. From the ODE model approximating these simulations, analytically derived ($\eta$, $T$) pairings where this equivalence holds are shown as neon-dashed curves overlaid on the heatmaps; along these curves, the effective growth rate corresponds to the value indicated by green-dashed frames on the colour bar. Note that the colour scales differ between heatmaps to clearly illustrate the impact of $\eta$ and

$T$ for a fixed dose and initial condition. Note further that in the 300 nM heatmap, all shown ($\eta$, $T$) pairings yield lower effective growth rates than the continuous treatment and thus no neon-dashed curves are seen. **c** The plots show simulated cell count dynamics in response to 1 and 4 cell updates per day, and treatment intervals of 1 and 4 days. Simulated mean counts (solid/dashed lines) and standard deviations (shaded bands) are shown for 100 simulation runs. These results demonstrate the benefit of targeting phenotypic adaptation in treatments by decreasing the number of cell updates per day. **d** We consider an initial condition in which all cells are in the most drug-resistant phenotype state ($x = 1$) in (**d**, **e**). **e** The experiments in (**b**) are repeated with the initial condition in (**d**). **f** The plots show simulated cell count dynamics in response to 1 and 4 cell updates per day, and treatment intervals of 1 day. Treatments commence after a delay period of 1 (left), 4 (middle), or 10 (right) days. Simulated mean counts (solid/dashed lines) and standard deviations (shaded bands) are shown for 100 simulation runs. The results demonstrate the benefit of leveraging phenotypic adaptation in treatments by delaying treatments and allowing drug-resistant cells to resensitize by traversing phenotype space.

---

thereby simultaneously linking phenotypic adaptation to both growth rates and genetic/transcriptomic mechanisms.

Whilst our results clearly indicate that phenotypic adaptation is directed towards states of high fitness in the regarded in vitro experiment, we can not determine if the semi-biased or biased update strategy best describes the regarded cells. This follows from the fact that we only consider mean growth rates in the model and do not have access to data that reveal population-level phenotype distributions. The observation that cell count data alone can not be used to distinguish between semi-biased and biased phenotypic adaptation opens up for a broader question: what type of experimental data is needed to identify phenotypic heterogeneity in cell populations and quantify how biased phenotypic adaptation is? We address this question in a subsequent study[31], where we have moved from a discrete to a continuous phenotype population model in order to allow for more rigorous mathematical analysis revealing that although cell count data alone is not sufficient to distinguish between the semi-biased and biased PU models, cell count data supplemented with information about birth and death events are. While we build our model on 3-day growth rates and (up to) four week treatment data, Russo et al.[32] combined long-term (over ten weeks) monitoring of experimental growth rates with an ordinary differential equation model to demonstrate that clinically approved targeted therapies induce drug-sensitive-to-tolerant adaptations in colorectal cancer cells. Further evidence of drug-induced cell plasticity was found by other recent studies combining mathematical modelling and in vitro experiments[33,34]. Beyond growth curve focussed studies, methods to quantify phenotype adaptation have been developed for data sets in which individual cells can be experimentally categorised into phenotype groups by, e.g., fluorescent markers[35,36].

Mathematical models, analysis and simulations, especially those integrated with data, have made significant contributions to our understanding of cancer dynamics and treatment responses[37,38]. To highlight a few contributions pertaining to BRAFV600E-mutant melanoma, Gerosa et al.[17] integrated proteomics and modelling to show that exposure to drugs that target the BRAFV600E-MEK-ERK pathway can cause non-genetic drug resistance by signal rewiring in the targeted pathway. Such rewiring was also identified as a mechanism of drug resistance by Fröhlich et al.[13], in an extension of the Gerosa et al. model, together with proteomic, transcriptomic and imaging data, to uncover that drug resistance can be mediated by the co-existence of two functionally distinct channels upstream of ERK, one initiated BRAFV600E monomers and the other by RAS. Whilst these models describe subcellular mechanisms that drive phenotypic adaptation, our model describes how cell-level phenotype traits, in the form of drug-dependent growth rates, change in response to BRAF-inhibitor exposure and removal. Comparing all three models, which complement each other, ours reveals less detailed subcellular information but is, on the other hand, considerably less data-intensive and therefore more accessible. Other data-integrated mathematical models of melanoma have been used to

identify dose combinations of BRAFV600E, MEK, and ERK inhibitors that yield synergistic treatment responses[39,40], predict initial treatment responses to BRAFV600E-inhibitors in xenografts[41], and demonstrate phenotypic plasticity and multi-stability[42]. Beyond melanoma, cell plasticity, and its impact on treatment responses and the evolution of therapy resistance, has been studied mathematically[12,43,44] through ordinary differential equation models[32] and partial integro-differential equation models[18,45–48] which, notably, can be used to suggest model-informed treatment strategies that impede the evolution of resistance.

Treatment scheduling and responses were also investigated in our current study. Our model-generated simulation results suggest that phenotypic adaptation can be both targeted and leveraged in an effort to suppress in vitro cell counts. These results support the development of molecular inhibitors that target mechanisms that drive cell plasticity[49]. Furthermore, strategies that leverage phenotypic adaptation, and, in other words, use cancers' ability to adapt against itself, can implicitly or explicitly be implemented through intermittent treatments and adaptive treatments[28,50]. In addition to these treatment strategies to combat adaptive resistance, there has been recent interest in epigenetic treatments that directly target phenotypic adaptation in the context of BRAF-inhibitor resistant melanoma[51,52]. A recent proof-of-concept clinical trial of combination epigenetic and MAPK inhibitors demonstrated durable clinical responses in a minority of participants[53]. We emphasise that the treatment-related results presented in this study are based on data-driven models of in vitro systems, and that appropriate model extensions, as well as in vivo or clinical data, are needed to draw any conclusions about the role of phenotypic adaptation in xenografts or clinical tumours. However, one of the major benefits of modelling in vitro systems to study cell plasticity is the access to temporal and easy-to-interpret data that reveal how cancer systems change over time. Such data are paramount to understanding dynamical aspects of cancer. Importantly, our limited understanding of cancer dynamics has been proposed as one of the main factors that hinder the development of efficient targeted therapy protocols[54,55] and whilst most cancer treatments target genomic cancer aberrations without consideration for evolutionary tumour aspects, clinical trials that are informed by mathematical models and account for tumour dynamics have started to emerge[56].

The growth rate-to-phenotype modelling pipeline presented in this work can be used to quantify phenotypic adaptation velocities, i.e., directions and rates, in any cell lines for which cell counts can be measured. Our pipeline can thus be used to estimate such velocities in cancerous, non-cancerous and mutated cell lines, and thereby quantitatively assess to what extent phenotypic adaptation stratifies cancer cells, mutated or not, from other cells. This functionality of our pipeline is of significant value to contemporary cancer research, as an emerging body of work has identified non-genetic resistance as a reason for drug resistance in multiple cancer types, including melanoma[7], neuroblastoma[57] and lung cancer[58]. One well-studied example mechanism of non-genetic resistance is the Epithelial-

Mesenchymal Transition (EMT)[8,49,59], although a multitude of other mechanisms have been reported[1,9,60–63]. Our growth rate-driven methodology is agnostic to these subcellular mechanisms and can therefore be used to directly examine phenotypic adaptation in its role as an emerging hallmark of cancer in melanoma and beyond.

## Methods

### Extracting growth rates from cell count data

The growth rate data used to map out the fitness matrix (Fig. 1) are extrapolated from cell count assays performed by Kavran et al.[9] who challenge WM239A-BRAFV600E melanoma cells with the BRAF-inhibitor encorafenib (LGX818) in vitro. In their experiments, cells are incubated with 500 nM encorafenib prior to a drug challenge with an encorafenib dose of 0, 0.1, 0.3, 1, 3, 10, 30, 100, 300, 1000, 3000, or 10000 nM. The data from the assays are reported in relative fold changes in cell counts. As such, for each drug concentration $d$, we have data for $FC_{72}^{\text{rel},d}$ which describes 'the fold change in cell counts after 72 h in response to dose $d$' relative to 'the fold change in cell counts after 72 h in response to dose 500 nM'. We use this data to extract $\lambda_d$, the effective growth rate for dose $d$ (in nM) between hours 0 and 72, through Eq. (1),

$$\lambda_d = \frac{\log(FC_{72}^{\text{rel},d})}{3} + \lambda_{500}, \tag{1}$$

where we have assumed exponential population growth over the 72 h period. The term $\lambda_{500} \approx 0.1059 \, \text{day}^{-1}$ is estimated using reported cell count data following a 7 day period of 500 nM treatment, applying linear regression on the log-scale under the assumption that growth is approximately exponential. The growth rate data are obtained using the code file `ProcessData.m` in our project's GitHub repository.

### Designing the fitness matrix

In this study, we model the net growth rate of a cell as a function of two variables: (1) the cell's phenotype state, and (2) the applied drug dose. The growth rates for cells in the melanoma system[9] that we study in this work are shown in the fitness matrix (Fig. 1d), in which matrix element $FM(x, d)$ describes the net growth rate for cells in phenotype state $x$ that are exposed to drug dose $d$. In this subsection, we describe how the fitness matrix is designed.

Based on the growth rate data, we say that drug-naive cells (0 wks in Fig. 1a) are in a drug-sensitive phenotype state that we arbitrarily denote by $x = 0$. Similarly, we say that cells that have been continuously exposed to 500 nM encorafenib for 1–4 weeks (1–4 wks in Fig. 1a) are in a drug-adapted phenotype state $x = 1$. Based on the data, we assume that cells can move from state $x = 0$ to state $x = 1$ within one week when exposed to 500 nM of the BRAF-inhibitor. We further make the simplifying modelling assumption that cells can revert from state $x = 1$ to state $x = 0$ following a drug holiday. This is an approximation that is based on experimental results for the studied melanoma system[9].

Assumption 1. Cells can revert from phenotype state $x = 1$ to phenotype state $x = 0$ and, by doing so, recover the growth rates of drug-naive cells.

Assumption 1 means that we can study phenotypic adaptation along a one-dimensional phenotype axis $x$ when modelling how cells adapt to exposure and removal of drug at dose $d$. We note that Assumption 1 may not be appropriate for all cell systems, as cell-level drug resistance in cancer ranges from reversible to irreversible[30]. Next, we make the simplifying modelling assumption that phenotype adaptation in directions of both increasing and decreasing $x$-values occurs in $n + 1$ discrete steps, where $n$ is the number of intermediate phenotype states between the extrema $x = 0$ and $x = 1$.

Assumption 2. Cells move between phenotype states $x = 0$ to $x = 1$, and back, in $n + 1$ discrete steps.

In the main body of our article, we choose to work with $n = 9$, in order to allow us to study density distributions of phenotype states, such as the ones depicted in Fig. 2c. However, our model could, for example, be reduced to study two-phenotype systems (with $n = 0$), or be extended to model continuous $x$-values, so that the phenotype density distributions would be described by partial differential equations. In the Supplementary Material (SM1), we include simulation results for a range of different $n$.

The growth rates $FM(x,d)$ for cells in states $x = 0$ and $x = 1$ are directly assigned from the growth rate data extrapolation described in the previous subsection. The process of assigning growth rates to intermediate states $x \in (0, 1)$ is described in the next subsection. Note that the net growth rate values $FM(x, d)$ in the right-most column of the fitness matrix ($x = 1$) are based on pre-incubation with 500 nM encorafenib, followed by 3 days of incubation in dose $d$. As such, the net growth rates of cells that have adapted to concentrations higher than 500 nM may have higher values than what our fitness matrix suggests. Experimental designs in which growth rates are measured following incubation with more than one dose (here 500 nM) would allow for a more detailed mapping of the fitness matrix.

### Modelling cell-level phenotypic adaptation with different update strategies

The four phenotype update (PU) strategies in Fig. 2a are formalised in this subsection. In order, these are the no, unbiased, semi-biased and biased PU strategies. We let $x_0$ denote the current phenotype state of an arbitrary cell. The probabilities of updating to higher ($x_+$) and lower ($x_-$) states, if possible with the restriction $x \in [0, 1]$, are given below.

$$PU1 : p(x_+) := 0, \\ p(x_-) := 0. \tag{2a}$$

$$PU2 : p(x_+) := \frac{\rho}{2}, \\ p(x_-) := \frac{\rho}{2}. \tag{2b}$$

$$PU3 : p(x_+) := \frac{\rho}{2} H\big( FM(x_+, d) - FM(x_0, d)\big), \\ p(x_-) := \frac{\rho}{2} H\big( FM(x_-, d) - FM(x_0, d)\big). \tag{2c}$$

$$PU4 : p(x_+) := H\big( FM(x_+, d) - FM(x_0, d)\big), \\ p(x_-) := H\big( FM(x_-, d) - FM(x_0, d)\big). \tag{2d}$$

Probabilities to remain at the current state $x_0$ are implied from the above equations through the relationship $p(x_+) + p(x_0) + p(x_-) = 1$. The value $FM(x,d)$ can be read from the fitness matrix in Fig. 1d, and denotes the net growth rate of a cell in phenotype state $x$ that is exposed to drug dose $d$. In Eqs. 2a–d, $H(\cdot)$ is the Heaviside function, and $\rho/2$ denotes the probability that a cell updates its phenotype state in a given direction at an update time (Fig. 2a). Note that the no update rule (PU1) can be considered a special case of both the unbiased (PU2) and semi-biased (PU3) update rule with $\rho = 0$. Furthermore, the biased update rule (PU4) can be regarded as the deterministic counterpart of the semi-biased update rule (PU3), in which the probabilistic pre-factor $\rho/2$ is omitted. That is, at each time step, PU3 permits fitness-improving transitions only with probability $\rho/2$, whereas PU4 enforces such transitions with probability one.

The four update strategies result in different phenotype density distributions. To demonstrate this visually, we plot these in response to 300 nM encorafenib exposure and removal following continuous and intermittent treatments (Fig. 4). While the non-directed strategies yield phenotype density distributions that only moderately change over time in response to intermittent treatments, the directed strategies yield phenotype distributions that markedly move between the drug-sensitive ($x = 0$) and drug-resistant ($x = 1$) extrema. Here, the simulation results have been produced with dose-dependent net growth rates from the fitness matrix (Fig. 1d), and

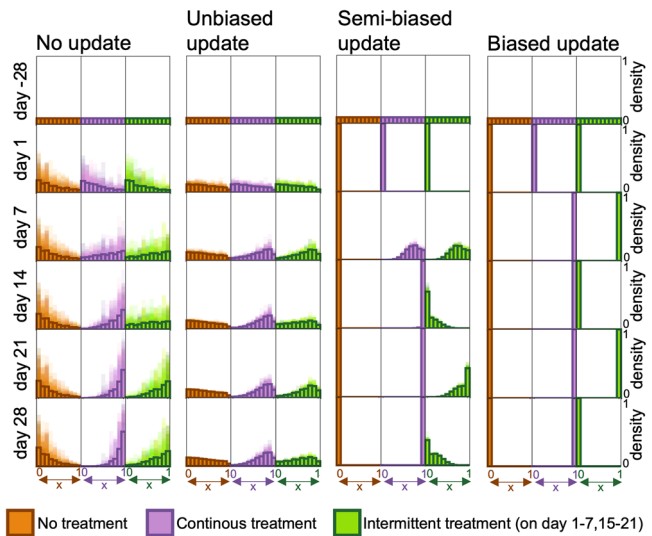

**Fig. 4 | The four candidate update strategies result in different dynamic phenotype density distributions.** The histograms show how cell population phenotype density distributions change over time in response to no, continuous and intermittent 300 nM BRAF-inhibitor treatments for the four update strategies described in Eqs. 2a–d. Densities are shown with 100 simulations layered over each other (shaded regions) and means (solid bars). 28 days before the experiments start (at day -28), cells are seeded with uniformly distributed phenotype states.

$\rho = 1$. The cells have a chance to update their phenotype state $\eta_{off} = 2$ and $\eta_{on=2}$ times per day in drug absence and presence, respectively, where these parameter values are used to ensure that cells can traverse phenotype space within a week (following observations in Fig. 1a).

### Implementing cell division, cell death and phenotype updates

The algorithm used to implement phenotype updates, cell division and cell death is illustrated in Fig. 2b and is outlined in Algorithm 1. After an initial cell seeding, we loop over all simulated time points $\hat{t}$ and cells $j$. At every simulated time point, the drug dose is set according to a treatment schedule. Next, it is decided whether cell division and death events should be resolved *before* (with probability 0.5) or *after* (with probability 0.5) the phenotype state update events. If a cell has a positive net growth rate, the rate corresponds to the probability per day that the cell divides and produces an identical daughter cell. If a cell has a negative net growth rate, the rate corresponds to the probability per day that the cell dies. Within a simulated time unit (in our case, day), all phenotype updates are performed consecutively.

**Algorithm 1**. **Implementation of cell division, cell death and phenotype updates**.

```
 1: Seed cells
 2: for t̂ = 1 to T do
 3:     set the drug dose d(t̂)
 4:     randomise a value v between 0 and 1
 5:     if v < 0.5 then
 6:         call function A
 7:         call function B
 8:     else
 9:         call function B
10:         call function A
11:     end if
12: end for
        function A: resolve cell division and cell death
13:     for j = 1 to J do
14:         g(x(j), d(t̂)) ← FM(x, d)
15:         if g(x(j), d(t̂)) > 0 then
```

```
16:             produce a daughter cell with a probability corresponding
                to g(x(j), d(t̂))
17:         else
18:             kill the cell with a probability corresponding
                to abs(g(x(j), d(t̂)))
19:         end if
20:     end for
        function B: resolve phenotype updates
21:     for j = 1 to J do
22:         for update = 1 to η do
23:             update x(j) according to the applied phenotype update rule
24:         end for
25: end for
```

### Using growth rates to parameterise phenotype and drug dose-dependent fitnesses

We linearly interpolate between the extrema $x = 0$ and $x = 1$ to assign dose-dependent growth rate values to intermediate phenotypes $x \in (0, 1)$ for all doses included in the fitness matrix (Fig. 1d). Since we do not have access to three-day fold changes in cell counts for 500 nM encorafenib, growth rates at this dose are instead estimated through a least-squares parameter fit (Eq. (3)) evaluated against fold change data for 500 nM continuous and intermittent treatments (Fig. 2d). In the fit, four free model parameters are estimated:

- FM(0, 500 nM): the growth rate for drug-naive cells in 500 nM encorafenib.
- FM(1, 500 nM): the growth rate for drug-adapted cells in 500 nM encorafenib.
- $\eta_{on}$: the number of times per day that cells have a chance to phenotype updates their phenotype when the drug is on.
- $\eta_{off}$: the number of times per day that cells have a chance to update their phenotype when the drug is off.

In biased update models, the directionality of the phenotype adaptation is dependent on the applied drug dose, as determined by the data. As such, cells move towards states of higher proliferative net fitness (Fig. 2a, c). To fit the free parameters, we further assume that the same phenotype update strategy applies both on and off drugs, although the phenotype update frequency may vary between drug-on and drug-off days[36].

Assumption 3. The same phenotype update strategy applies both on and off drugs, and the update rates may differ between drug-on and drug-off days.

The parameter combination $C_\ell^*$ that minimises the sum of squared distances between mean data values and model simulations (for 100 simulation runs) for continuous and intermittent treatments, simultaneously, are obtained for all phenotype update strategies $\ell$.

Thus, for each update rule $\ell$, the four selected parameter values

$$C_\ell^* = \left( FM(0, 500\ nM), FM(1, 500\ nM), \eta_{on}, \eta_{off} \right)$$

that simultaneously minimise the squared distance between the mean model predictions $M_{s,t}$ and in vitro data $D_{s,t}$ are obtained through

$$C_\ell^* = \arg\min_{C_\ell} \sum_{s \in \{cont,\ int\}} \sum_{t=1}^{4} \left( M_{s,t}(C_\ell) - D_{s,t} \right)^2, \quad (3)$$

implemented in MATLAB. In Eq. 2, the model-to-data distances are evaluated for the settings continuous and intermittent 500 nM treatments ($s \in \{cont, int\}$) at four time points $t$ (excluding the initial condition). The fit is bounded by growth rates for the fitness matrix-adjacent drug doses 300 and 1000 nM between which 10 linearly interpolated values are tested. Further, $\eta_{on}$ and $\eta_{off}$ can take values $1/h$ and $k$, for integers $h$ between 2 and 10, and $k$ between 0 and 10. An update frequency of $1/h$ means that

phenotype updates occur every $h$th day, and an update frequency of $k$ means that phenotype updates occur $k$ times a day.

The parameter combinations $C_\ell^*$ inferred through Eq. (3) are used to produce the simulation results in Fig. 2d, f for the 500 nM sub-panels, where

$$C_1^* = \left(-0.202 \text{ day}^{-1}, \, 0.171 \text{ day}^{-1}, \, 0 \text{ updates day}^{-1}, \, 0 \text{ updates day}^{-1}\right),$$

$$C_2^* = \left(-0.202 \text{ day}^{-1}, \, 0.171 \text{ day}^{-1}, \, 1 \text{updates day}^{-1}, \, 3 \text{ updates day}^{-1},\right),$$

$$C_3^* = \left(-0.335 \text{ day}^{-1}, \, 0.146 \text{ day}^{-1}, \, 3 \text{ updates day}^{-1}, \, 4 \text{ updates day}^{-1},\right),$$

$$C_4^* = \left(-0.268 \text{ day}^{-1}, \, 0.171 \text{ day}^{-1}, \, 1 \text{ updates day}^{-1}, \, 3 \text{ updates day}^{-1},\right).$$

As is shown in the Supplementary Material (SM1), fitting $\eta_{\text{on}}$ and $\eta_{\text{off}}$ only, using M(0, 500 nM) and FM(1, 500 nM) obtained from log-linear interpolation between 300 and 1000 nM growth rates, does not result in simultaneously data-matched simulations for continuous and intermittent treatments. For all simulations in the main part of this article, $\rho$ is fixed at $\rho = 1$ and thus the values of $\eta_{\text{on}}$ and $\eta_{\text{off}}$ determine the speed of phenotype adaptation, given a specified $n$. In the Supplementary Material (SM1), we have re-fitted $C_\ell^*$ for variations of $\rho$. As expected, these supplementary results demonstrate a non-linear trade-off relation between $\rho$, $n$, and $\eta_{\text{on,off}}$; if the probability to accept update propositions (i.e., $\rho$) decreases, then similar macroscopic system behaviour can be recovered by altering the number of times that the cells can update their phenotype states per day, or the number of phenotype states $n$ between the extrema $x = 0$ and $x = 1$. The observation that the directed phenotype update models better capture dynamic cell count data than the undirected models for $n > 0$ is robust to the investigated variations of $\rho \in \{0.25, 0.50, 0.75, 1\}$ and $n \in \{0, 1, 3, 5, 7, 9\}$.

**Deriving analytical expressions to identify simulated intermittent treatment regimes that outperform their continuous counterparts**

To predict when intermittent treatment schedules outperform continuous schedules, we derive analytical expressions for long-term effective growth rates under the biased phenotype update strategy. We let $t_{\text{on}}$ denote the duration (per cycle) that the drug is *on*, and $t_{\text{off}}$ denote the duration (per cycle) that the drug is *off*. $\omega$ is the phenotype adaptation velocity (in both directions) such that $\eta = \omega/\Delta$ gives the number of phenotype updates per day. In the Fig. 1d example, $\Delta = 0.1$ is the step size arising from the choice $n = 9$. Implicit to the biased update rule is that, at any time in a simulation, there is no phenotypic heterogeneity between cells (Fig. 2c). As such, all cells have an identical phenotype, denoted by $x(t)$, the dynamics of which are governed by

$$\frac{dx}{dt} = \begin{cases} \omega, & x < 1 \text{ and the drug is on}, \\ -\omega, & x > 0 \text{ and the drug is off}, \\ 0, & \text{otherwise}, \end{cases} \quad (4)$$

for drug doses 300 and 500 nM, and any other doses listed inside the top arrow of Fig. 2e. Now, let's assume that we have a cell population in equilibrium such that $x(s) = x(0)$. Then, over a time period $\tau = t_{\text{on}} + t_{\text{off}}$, the cell count $N(s + \tau)$ for cells on intermittent treatment is given by

$$N(s + \tau) = N(s)e^{\lambda_{\text{eff}} \cdot \tau}, \quad (5)$$

where the effective growth rate, $\lambda_{\text{eff}}$, is

$$\lambda_{\text{eff}} = \frac{1}{\tau}\left(\int_0^{t_{\text{on}}} \text{FM}(x(t), d)\, dt + \int_{t_{\text{on}}}^{\tau} \text{FM}(x(t), 0)\, dt\right). \quad (6)$$

Solving Eq. (6) yields the effective growth rate for cells on intermittent treatment to be

$$\lambda_{\text{eff}} = \begin{cases} \frac{\Delta\text{FM}(0)-\Delta\text{FM}(D)}{2\tau\omega} + \frac{t_{\text{on}}\text{FM}(1,D)+t_{\text{off}}\text{FM}(0,0)}{\tau}, & \text{if } t_{\text{on}}, t_{\text{off}} > \frac{1}{\omega}, \\[6pt] \frac{\omega t_{\text{on}}^2(\Delta\text{FM}(0)+\Delta\text{FM}(D))}{2\tau} + \frac{t_{\text{on}}\text{FM}(0,D)+t_{\text{off}}\text{FM}(0,0)}{\tau}, & \text{if } t_{\text{on}} < t_{\text{off}} \leq \frac{1}{\omega}, \\[6pt] \frac{-\omega t_{\text{off}}^2(\Delta\text{FM}(0)+\Delta\text{FM}(1))}{2\tau} + \frac{t_{\text{on}}\text{FM}(1,D)+t_{\text{off}}\text{FM}(1,0)}{\tau}, & \text{if } t_{\text{off}} < t_{\text{on}} \leq \frac{1}{\omega}, \\[6pt] \frac{\text{FM}(0,0)+\text{FM}(0,D)+\text{FM}(1,0)+\text{FM}(1,D)}{4}, & \text{if } t_{\text{on}} = t_{\text{off}} = \frac{1}{\omega}, \\[6pt] \frac{\omega t_{on}(\Delta\text{FM}(0)+\Delta\text{FM}(D))}{4} + \frac{\text{FM}(0,0)+\text{FM}(0,D)}{2}, & \text{if } t_{\text{on}} = t_{\text{off}} < \frac{1}{\omega}, \\[6pt] \frac{\Delta\text{FM}(0)-\Delta\text{FM}(D)}{4t_{on}\omega} + \frac{\text{FM}(0,0)+\text{FM}(1,D)}{2}, & \text{if } t_{\text{on}} = t_{\text{off}} > \frac{1}{\omega}, \end{cases} \quad (7)$$

where

$$\Delta\text{FM}(d) = \text{FM}(1, d) - \text{FM}(0, d), \quad (8)$$

and $D$ is a chosen drug dose such that Eq. (4) holds. The expressions in Eq. (7) are derived as shown in the Supplementary Material (SM2), and allow a user to find $\lambda_{\text{eff}}$ for customised values of $D, t_{\text{on}}, t_{\text{off}}, \omega$. To, next, draw fair comparisons between cells on continuous and intermittent treatments, we can compare the long-term effective growth rate of cells under both regimes. To avoid effects of the initial drug desensitisation for cells undergoing continuous treatments, we consider the effective growth rates under continuous treatment to be given by FM(1, $D$). Thus, we expect intermittent treatment to outperform continuous treatment if $\lambda_{\text{eff}}(D, t_{\text{on}}, t_{\text{off}}, \omega)$ is lower than FM(1, $D$). In Fig. 3, we consider a treatment regime with equal on and off drug intervals, such that $t_{\text{on}} = t_{\text{off}} = \tau/2 = T$. Results are produced with net growth rates obtained through log-linear interpolation between FM($x$, 300 nM) and FM($x$, 1000 nM) so that FM(0,500 nM) $= -0.287$ day$^{-1}$ and FM(1,500 nM) $= 0.074$ day$^{-1}$ (see Supplementary Material, SM1). Inserting this choice into Eq. (7) for $T\omega \geq 1$, we find that intermittent treatment will outperform continuous treatment if $T\omega < 6.75245$ or, equivalently, $T\eta < 67.5245$.

We remark that Eq. 4 can be modified to investigate scenarios where phenotype adaptation occurs at different rates during drug-on and drug-off periods, by allowing the drug-off rate to differ from the negative of the drug-on rate (further, these rates need not be constants). Similarly, Eq. 6 can be modified to model intermittent treatment schedules with on/off proportions other than 1:1, by altering the integration limits. Thus, the ODE framework presented in Eqs. 4–6 can be readily adapted to study a broad range of phenotype adaptation and treatment settings. For such modifications, deriving new effective growth rates $\lambda_{\text{eff}}$ is straightforward using the procedure outlined in the Supplementary Material (SM2).

**Statistics and reproducibility**

The simulation result summary statistics presented in this article are mean values and standard deviations from 100 simulation runs, as indicated by the figure legends. The simulation and analysis results presented in this study are reproducible using the code files available on the public GitHub repository https://github.com/SJHamis/phenotype_adaptation.

**Reporting summary**

Further information on research design is available in the Nature Portfolio Reporting Summary linked to this article.

**Data availability**

The simulation data produced in this study, as well as the in vitro data used in this study, are available on the public GitHub repository https://github.com/SJHamis/phenotype_adaptation.

**Code availability**

The model is implemented in MATLAB. Information on how to access, run and modify the code files is available on the public GitHub repository

https://github.com/SJHamis/phenotype_adaptation. The parameter values used to generate the simulation results presented in this article are tabulated in the Supplementary Material (SM3).

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

## Acknowledgements

SH was funded by the Swedish Research Council (project 2024-05621), Wenner-Gren Stiftelserna/the Wenner-Gren Foundations (WGF2022-0044), and the Kjell och Märta Beijer Foundation. A.P.B. thanks the Mathematical Institute, Oxford for a Hooke Research Fellowship. A.L.J. thanks the London Mathematical Society. This project was partially supported by the European Union's Horizon 2020 research and innovation programme under the Marie Skłodowska-Curie grant agreement No 945298-ParisRegionFP (C.V.). C.V. is a Fellow of the Paris Region Fellowship Programme, supported by the Paris Region. This work was partially supported by a Heilbronn Institute for Mathematical Research Small Maths Grant to T.C.

## Author contributions

S.H., A.P.B., A.L.J., C.V., P.K.M., and T.C.—Conceptualisation; S.H., A.P.B., A.L.J., C.V., P.K.M., and T.C.—Methodology; S.H., A.P.B., A.L.J., C.V., P.K.M., and T.C.—Writing (review and editing); S.H., A.P.B., and T.C.—Writing (original draft); S.H., A.P.B., and T.C.—Formal analysis; S.H. and A.P.B.—Investigation; S.H., A.P.B.—Software; S.H.—Visualisation.

## Funding

## Competing interests

The authors declare no competing interests.
