## [Transparent Peer Review file · Communications Biology]

Growth rate-driven modelling suggests that phenotypic adaptation drives drug resistance in BRAFV600E-mutant melanoma

Corresponding Author: Dr Sara Hamis

Version 0:

Reviewer comments:

Reviewer #1

(Remarks to the Author)

Please see the attached document.

Reviewer #2

(Remarks to the Author)

Growth rate-driven modelling reveals how phenotypic adaptation drives drug resistance in BRAFV600E-mutant melanoma

Sara Hamis¹✉, Alexander P Browning², Adrienne L Jenner³, Chiara Villa⁴, Philip K Maini², Tyler Cassidy⁵.

Overview and Major Claims:

This well written and clear paper presents a mathematical model of melanoma adaptation and the authors use it to explore response to treatment. This work focuses specifically on mutations in BRAFV600E and treatment with the BRAF-inhibitor encorafenib. The paper focuses on multi-state phenotypic model with various methods of phenotype updating. The authors compare different undirected (none, unbiased) and directed (semi-biased, biased) modes of phenotypic evolution. These models of resistance simulate phenotypic plasticity (unbiased, semi-biased, biased updating) in addition to resistance arising from pre-existing heterogeneity (no updating). The authors present the first use of this model type for the interpretation of experimental data. The authors show that in their model, directed adaptation is required to reproduce experimental observations. Additionally, they claim that it provides a growth rate data driven explanation for why intermittent treatments outperform continuous treatments at 500 nM encorafenib. They go on to explore the extent to which intermittent treatment regimes outperform continuous maximal dosing for the directed phenotypic update model of resistance. The discussion, methodology, and results are clear, with minor updates needed for clarity and context. While the model is somewhat agnostic to biological mechanism, the authors discuss melanoma specific biology to appropriately contextualise their findings.

Novelty: The novelty of this work lies in the combination of phenotypic state modeling with experimental data in melanoma. The authors also present novel work in the further use of their fitted model in making predictions under clinically relevant treatment schedules and treatment holidays. They present interesting findings around the non-monotonic relationships between adaptive behavior and treatment scheduling.

Convincing: Yes, except Fig 3c.

Concerns (Fig 3.)

This figure is a little difficult to parse, perhaps due to missing visual schema on dosing and empty boxes and because initial condition subfigures seem to be a little excessive/add clutter. Green dashed lines are under-explained. In c) and f) It is unclear why there are transparent boxes for 0nM and 500nM encorafenib. The legend does not mention 0nM. Perhaps there is meant to be a colored intermittent dosing regime not shown. Importantly, in Fig. 3c) The n=4 updates (dashed) per day has a lower cell count than n=1 (solid), this seems to oppose to the claim in the text (that treatment is more effective when

updates per day are reduced).

Concerns (Minor - Clarity of presentation)

Fig 1. d) There is a mix of significant figures and decimal places, consistency (and increased text size) would be beneficial in the presentation of this subfigure. Bolded yellow on gray is also difficult to read and the authors may wish to choose something like underlining the negative growth rates instead.

Fig 3. b) The legend and main text would benefit from rephrasing and explanation about the analytical result, and at least a direction to the analytical results which have been placed in the methods. The authors may wish to highlight the lack of a boundary at 300nM.

Concerns (Minor - Discussion of experimental choices and limitations):

The authors use very specific parameters including numbers of states and assumptions (linear interpolation between states, uniform seeding). While these assumptions are often discussed, emphasis on the choice (literature, fitted optimal or arbitrary) of key parameters or parameter ranges when they arise would improve manuscript clarity. (Primarily because fitted parameters are Results themselves).

The authors neglect to mention (even in discussion) how their explanation for intermittent therapy efficacy fits into the literature that explains intermittent schedule success via ecological/frequency dependent effects. Specifically so called "adaptive therapy" can provide alternate data-driven explanations for the success of intermittent therapies (see adaptive therapy literature and models).

Validity of Statistical Approach: Mathematical model appropriately constructed and defined.

Ease of Reproduction:

Appropriate level of detail included for figures. Code provided (github).

Best,
Rowan Barker-Clarke, PhD.

Reviewer #3

(Remarks to the Author)

The paper, "Growth rate-driven modelling reveals how phenotypic adaptation drives drug resistance in BRAFV600E-mutant melanoma," provides a framework for understanding the role of phenotypic adaptation in cancer drug resistance. Through a computational model informed by experimental data, the authors explore how melanoma cells adapt their phenotypes in response to BRAF inhibitor treatments. Their model incorporates discrete, plastic phenotype states that dynamically respond to drug exposure, revealing that adaptation is directed towards high-fitness states. This mechanism explains the observed efficacy of intermittent treatment schedules over continuous dosing, offering a new perspective on optimizing therapy strategies. The integration of growth rate data and simulation experiments showcases the model's robustness and potential translational value.

In general, I like the paper and would like to ask the authors to provide some explanation of how they generate the Fitness Matrix.

Version 1:

Reviewer comments:

Reviewer #1

(Remarks to the Author)
Please see attached review.

Reviewer #2

(Remarks to the Author)
The authors made very responsive improvements, and their clarity around the modeling approach strengthens a high-quality manuscript.

Copyediting comment only:

Line 34: by epigenetic changes in cancer cells,
should read by **epigenetic** changes in cancer cells,

Reviewer #4

(Remarks to the Author)

Please see attached comments for a detailed discussion.

Version 2:

Reviewer comments:

Reviewer #1

(Remarks to the Author)

Response to reviewer 1

In this work, the authors seek to explain an experimental finding that intermittent treatment outperforms continuous treatment in BRAF-mutated melanoma using mathematical modeling. They propose a model involving several cell states, where cells transition progressively from an initially drug-sensitive state to an eventually drug-adapted state. Simple alternative hypotheses of the nature of this transition are tested, and it is shown that the experimental data is most consistent with a model which involves one-way (semi-biased or biased) transitions from drug-sensitivity to drug-adaptation in the presence of drug and one-way transitions back in the absence of drug. The authors conclude that phenotypic adaptation towards higher fitness explains why intermittent treatment outperforms continuous treatment, and they discuss how phenotypic adaptation can potentially be targeted and leveraged in treatment protocols.

This work is timely, as it considers the important concepts of phenotypic plasticity and drug-induced adaptation, and it explores how mathematical modeling can be used to understand them. While it's not necessarily surprising that a preference for intermittent treatment emerges under one-way transitions to a drug-adapted state in the presence of drug and one-way transitions back in the absence of drug, it is interesting that the experimental data can be fit well with such a simple model. My main concern with the work is that its scope and applicability appear limited. As far as I can tell, the work mostly produces a qualitative explanation of an experimentally known fact, that a 0/500nM intermittent treatment outperforms a 500 nM continuous treatment. The framework as currently formulated does not enable analysis of intermittent or continuous treatment strategies over all possible different doses (which I argue is necessary to tackle the general question of whether intermittent treatment is preferred to continuous treatment), and it does not enable prediction of whether an intermittent treatment outperforms a continuous treatment from multiple-dose growth rate data alone. In addition, some modeling choices are made which I think deserve further explanation or exploration. These points are outlined in more detail in the comments below.

Thank you for a detailed review with helpful insights, suggestions, and references. We feel that this constructive feedback has helped us improve the quality of our manuscript. We describe how we address these main comments in points 1-7 below.

Reviewer 1 - main comments

Point 1. I have some doubts about the derivation of the fitness matrix in Figure 1. The growth rates shown in (a) are obtained using cells that have been incubated in 500 nM encorafenib and then rechallenged with the drug. Unfortunately, the experimental data does not include drug rechallenge at 500 nM, but it is probably not a coincidence that only doses larger than 500 nM are able to kill the cells. It seems plausible that cells exposed to 1000, 3000 or 10000 nM will adapt the ability to survive or at least persist at those doses and only be killed at larger doses. In other words, the drug-adapted phenotype that emerges under > 500 nM may be different from the one that emerges under 500 nM and its dose-response curve may be different. I also wonder whether it is necessary to derive an entire fitness matrix if the focus of most subsequent analysis is on the dynamics under 500 nM.

To address these concerns regarding the fitness matrix (specifically the part for encorafenib doses over 500 nM), we have now added a new methods subsection that describes how the fitness matrix is derived. The subsection is called: **“Designing the fitness matrix”**.

In this subsection, we have taken care to clearly state that cells may adapt to become even more drug-addicted if exposed to doses higher than 500 nM. As part of the new subsection we write:

“Note that the net growth rate values $FM(x, d)$ in the right-most column of the fitness matrix ($x = 1$) are based on pre-incubation with 500 nM encorafenib, followed by 3 days of incubation in dose d . As such, the net growth rates of cells that have adapted to concentrations higher than 500 nM may have higher values than what our fitness matrix suggests.”

While we have decided to keep the full fitness matrix in Figure 1d, we hope that the new subsection resolves these concerns on the matter.

Point 2. In Figure 3, the results for 300 and 500 nM encorafenib are quite distinct, which is based on log-linear interpolation of the growth rate under 500 nM between experimental observations under 300 and 1000 nM. As in the previous comment, I am not sure how confidently this interpolation can be made for the drug-adapted cells. In Figure 1(a), there is an overall trend upwards in growth rate for drug-adapted cells from no drug (0 nM) to 300 nM drug exposure. Since the cells have been incubated in 500 nM, it is reasonable to assume that 500 nM is a critical dose in this context, and that the growth rate may continue to increase from 300 nM to 500 nM or at least stay the same. The experimental data seems insufficient to determine where exactly the turning point is and it is therefore not possible to say with great confidence that results differ between 300 and 500 nM. Also note that for the results in Figure 2, the growth rate of drug-adapted cells under 500 nM is fit to data, and the result is that the no update, unbiased update and biased update models all prefer to set the growth rate under 500 nM exposure equal to the growth rate under 300 nM exposure, which is consistent with the above reasoning. I suggest that the authors rethink this section or at least address the uncertainty.

We agree with the reviewer that we cannot say with great certainty how much the net growth rates in response to 300 nM or 500 nM encorafenib differ from each other. This follows from the fact that we do not have access to 3-day growth-rate data for cells under 500 nM, but rather interpolate between adjacent drug dose-dependent growth rates to find it.

To address these concerns, we have edited the main manuscript (Result section) and added two Supplementary Material figures (S2,S3):

- We write: “For all update models, with the exception of the biased update model, cell counts in response to continuous and intermittent treatments are similar in response to 300 nM and 500 nM encorafenib. This result is further discussed and visualised in the Supplementary Material (Figure S1). The Supplementary Material (Figure S3) also includes simulated cell counts in response treatments for different choices of phenotype granularity, i.e., n , showing that if we consider the cell population to be phenotypically non-binary, then the directed strategies better explain the observed data than do the non-directed strategies.” *Please see the Result section, line 161.*
- **Supplementary Figure S2:** Includes plots comparing simulated cell counts in response to treatments with 300 nM, 500 nM (fitted), and 500 nM (using growth rates obtained through log-linear interpolation).
- **Supplementary Figure S3:** Includes plots for simulated cell counts in response to 500 nM treatments with different phenotype granularity (i.e., values of n).

We would like to emphasise that our main objective in the second result section "Phenotypic adaptation in BRAFV600E-mutant melanoma is directed towards high fitness" and Figure 2 (with the same name) is to select between phenotype update models (as depicted in Figure 2a). For this objective, the differences in response to 300 nM and 500 nM doses are not a direct problem, in fact Figure 2d show similar behavior for treatment responses to these doses. We do, however, hope that the bullet points above have addressed these concerns.

In the main text, we have chosen to keep Figure 2d as is (especially the 500 nM treatment response curves, inserted below). The reason for this is that the 500 nM plot allows for simple comparison with the Kavran et al. (Kavran, 2022) paper (especially Figure 2b in their paper, which is inserted below). With these axes ranges, the 300 nM growth curves do not fit into the plot window for the biased phenotype update model.

Left: Bottom-part of Figure 2d in our manuscript. Right: Figure 2B in Kavran et al. (Kavran, 2022).

Point 3. In the introduction, it is stated that the framework can identify when intermittent treatments may outperform continuous treatments using multiple-dose growth rate data. As far as I can tell, the current approach is only able to explain a preference for intermittent treatment by fitting to experimental data involving that treatment. A framework has not been suggested that enables prediction of whether an intermittent treatment outperforms a continuous treatment using multiple growth rate data alone. In addition, the general question of whether intermittent treatment outperforms continuous treatment would require investigating all possible dose values for each type of strategy. The modeling framework as currently formulated is not set up to answer this question, since there should presumably be some dose dependence in the phenotypic transition parameters ρ and η . In addition, following the previous comments, even though the experimental data involves drug challenge at multiple doses, it is unclear to me how much can be said about the drug response dynamics outside 500 nM. This would significantly limit the scope of the work and the applicability of this approach for evaluation of dosing strategies.

We appreciate these concerns and have accordingly edited the following sentence in the introduction.

- **Old text:** "Accordingly, our framework can identify when intermittent treatments may be more effective than continuous treatments directly from multiple-dose growth rate data from untreated and treated cells."
- **New text:** Based on multiple-dose growth rate data from untreated and treated cells, our framework thus provides a mechanistic explanation for when and why intermittent treatments may be more effective than continuous treatments.

With the available data, we can only experimentally confirm that intermittent treatments outperform continuous treatments for 500 nM, 1 week on/1 week off treatments for 4

weeks (Figure 1d, bottom row). This result is, as the reviewer points out, indeed known from experiments prior to our modelling work. In response to these remarks, please note that our results section are structured as follows:

- In the first results section, our main contribution is to introduce a method to map out a fitness matrix from growth rate data (Figure 1d). We specifically apply our method to Kavran et al.'s (2022) melanoma system, but fitness matrices could be extracted from other cell systems.
- In the second results section, our main contribution is to infer that phenotypic adaptation in the regarded melanoma cell system is directed towards high fitness states (i.e., that the directed phenotype update strategies apply). The same investigation could be applied to other cell systems to find out which phenotype updates rules apply there.
- In the third results section, we combine these results (the fitness matrix and the inferred directionality of the phenotype updates) to make predictions about when intermittent treatments can outperform continuous treatments in the melanoma system, given the number of phenotype updates per day (η), and the treatment schedule (T). *This is where we argue that we can “identify when intermittent treatments may outperform continuous treatments”.* While we do not have enough experimental data to confirm these results, we have simulation results and analytical results that are in agreement (Figure 3 b,e).

Point 4. While I appreciate the desire to test simple potential alternatives of the model dynamics, I have the following questions and thoughts on the models:

i) Is the distinction between the semi-biased and biased update strategies important? First, the semi-biased strategy in its general form includes the biased strategy as a special case when the probability of remaining in the current state is zero. Second, I would think that it matters less for the outcome whether the dynamics are semi-biased or biased and more what the ultimate transition rate towards resistance is ($\eta\rho$). I can also imagine that with limited data, the parameter $\eta\rho$ may be more identifiable than η and ρ individually. I suggest that the authors think about reformulating the semi-biased/biased options as a single biased option and allowing the update probability ρ to be free when fitting to the data. I also suggest that the authors think about whether a formulation that focuses on the transition rate makes more sense.

We understand this comment and agree with the reviewer that it is, indeed, true that some of our phenotype update (PU) models can be viewed as “special cases” of other PU models. In line with what the reviewer notes, we now write: **Note that the no update rule can be considered a special case of both the unbiased and semi-biased update rule with $\rho = 0$. Further, the biased update rule can be considered a special case of the semi-biased update rule with $\rho/2 = 1$. Please see the Methods section, line 459.**

We have chosen to keep all four PU models and motivate why in the text. We write **“Biologically, the unbiased PU strategy corresponds to phenotype instability, while the biased PU strategy corresponds to stress-induced phenotypic adaptation, in accordance with work by Chisholm et al.¹ Ultimately, the mechanisms responsible for the development of resistance are not yet fully known, and we therefore also explore the more general semi-biased PU strategy than encompasses both phenotypic instability and adaptation.” Please see the Result section, line 124.**

Regarding these comments on compounding η and ρ (rather than viewing them separately): we could have done this, and it would have reduced a parameter. Since we have fixed ρ , and are only fitting η , we are in practice doing this, which we now emphasise in the main manuscript and write: “For all simulations in this article, ρ is fixed at $\rho = 1$ and thus the values of η_{on} and η_{off} determine the speed of phenotype adaptation.” Please see the Methods section, line 437.

We have, however, chosen to create and work with a generalisable framework that makes it possible to separate the questions: “how often can phenotype updates happen (what is η), and how likely is it that a proposed step is taken (what is ρ).

Regarding “I also suggest that the authors think about whether a formulation that focuses on the transition rate makes more sense.” Since we fix ρ , we are essentially focussing on the transition rate (please see the most recent red text snippet).

Point 5. In the introduction, the model is stated as having an arbitrary number of states S , yet in the main text it is assumed that cells evolve from drug-sensitivity to drug-adaptation through 10 steps. Is there evidence for multi-stage adaptation and in particular a 10-step process? I wonder for example whether a singlestage model with semi-biased updating and the correct values of ρ and η can reproduce a gradual move of the population from mostly drug-sensitivity to mostly drug-adapted cells and also explain the experimental results. Given that this is a debated question, put in focus for example in [1], I think it should be made clear whether it is possible to distinguish single vs. multistage evolution from the data and whether the choice of the number of stages affects the relative preferability of the candidate models.

To address your comment on why we assume multi-step phenotype adaptation, we have added the following text in the manuscript:

“The multi-step adaptation model is based on the observation that protein expression for L1CAM, a marker for drug resistance in melanoma, changes gradually in response to external pressures in the regarded experiments (Kavran, 2022). However, in the supplementary material we repeat the key simulation experiments performed in this study for other values of n , including $n = 0$, resulting in a single-step adaptation model.” Please see the Result section, line 94.

We have also included a new figure (S3) in the supplementary material where we include simulated cell counts over time in response to treatments for different choices of phenotype-granularity (n), please see our response to the reviewer’s second remark.

Point 6. I would be interested in seeing some exploration of models where there are two-way transitions between adjacent states without the transitions being unbiased. There are many experimental works where drug-sensitive and drug persistent cells coexist before treatment (for example [2, 3]), despite the drug persistent cells having a significant fitness disadvantage off-drug, and the transition rates between them are generally asymmetric [4]. I wonder if it is possible to explain the dynamics with a two-way transition model, where transitions towards the drug-sensitive state are faster in the absence of drug, but transitions towards the drug-adapted state become elevated under the drug. This would preserve heterogeneity in the population at any time, which I think is more reasonable than all cells existing in the same state at any time, and it would involve clear quantification of the drug effect. In my opinion, this is a reasonable default model which should at least be discussed.

From what we understand, the reviewer is asking about a phenotype update model such that cells

- move fast(er) towards $x=0$ (the drug-sensitive state) when the drug is off,
- move slow(er) towards $x=1$ (the drug adapted state) when the drug is on,

and the model is “without the transitions being unbiased”, i.e., with the transitions being biased. This update speed can be tuned by the parameters η_{on} (the number of updates when the drug is on) and η_{off} (the number of updates when the drug is off). We do this fitting to produce Figure 2d. Please see the methods section “Using growth rates to parameterise phenotype and drug dose-dependent fitnesses” for details, and the newly added Figure S3 in the Supplementary Material.

Point 7. Have the authors considered asymmetric on/off intermittent strategies? For example, in [5] which considers BRAF-mutated melanoma, the authors find a preference of a 2:1 on/off strategy compared to a continuous strategy.

To make our work more generalisable, in line with the reviewer’s comment, we have included a Wolfram Mathematica notebook that lets users derive effective growth rates for any choice of treatment on and treatment off durations. The notebook is included in **Supplementary Material SM2**.

We have, accordingly, edited the subsection “Deriving analytical expressions to identify simulated intermittent treatment regimes that outperform their continuous counterparts” to not require that the *treatment on* time is the same as the *treatment off* time. As such, the analysis now encompasses 1:1, 2:1, and other repeating on/of treatment schedules. *Please see especially equation 6 in the Methods section.*

Point 8. The discussion of targeting phenotype adaptation on page 3 is a bit vague. There already exist epigenetic treatments which target non-genetic mechanisms. Can the authors, either on page 3 or in the discussion section, contemplate what kind of currently available epigenetic treatments could be used for this purpose?

Approaches to target phenotypic adaptation is a very important question in the development of new treatments for melanoma. Here, we considered an existing small molecule that targets a common driver mutation in BRAF that leads to constitutive activation of the MAPK signalling pathway. We considered drug-resistance arising from phenotypic adaptations, which may result from epigenetic changes. In this context, treatment strategies targeting this phenotypic adaptation are primarily to scheduling changes, such as the intermittent dosing explored by (Kavran 2022), or other strategies, like *adaptive therapy*, with existing (and approved) therapies, like encorafenib, as we are able to directly compare our model against preclinical data in these cases.

However, as the reviewer mentioned, there have been recent advances in epigenetic targeting therapies, and both (Emmons, 2019; Gallagher, 2018) studied the impact of HDAC8 inhibition in BRAF mutant melanoma cells resistant to the first-gen BRAF inhibitors in preclinical experiments. We have updated the introduction to discuss these recent developments. However, as we have not considered these epigenetic treatments in any of our modelling, we are reluctant to speculate on which precise mechanisms may be targeted. We have added a brief mention of these treatment approaches to the discussion. We write: “In addition to these treatment strategies to combat adaptive resistance, there has been recent interest in epigenetic treatments that directly target phenotypic adaptation in the context of BRAF inhibitor resistant melanoma.^{2,3} A recent proof-of-concept clinical trial of combination epigenetic and MAPK inhibitors demonstrated durable clinical responses in a minority of participants.⁴” *Please see the Discussion, line 313.*

Point 9. In general, I think that the review and discussion of previous work in the area of mathematical modeling of phenotypic plasticity and drug-induced adaptation, as well as non-genetic treatments, is incomplete. For example:

- Pertaining to the discussion on page 7 of what type of experimental data is needed to identify phenotypic heterogeneity in cell populations and quantify how biased phenotypic adaptation is, there is already some work on this question. First, the works [6] and [4] and related subsequent work have covered the possibility of quantifying phenotypic transitions when it is possible to distinguish and sort cells in vitro using surface markers. This approach could presumably be used to quantify how biased phenotypic adaptation is in vitro. I think there also exist lineage-tracing methods which I'm less familiar with. However, even if only total cell count data is available, some things can still be said. For example, I think it's important to discuss [7], which infers a relationship between the transition rate from drug-sensitivity to drug-persistence and drug dose using total cell count data. This work uses the fact that if the population is observed over a sufficiently long time, the drug-sensitive and drug-adapted phenotypes can be distinguished from total cell count data by comparing the initial decay rate under drug-exposure with the longer-term growth rate. [8] and [9] are more recent preprints which address a similar question.

We have now expanded our discussion on how mathematical modelling and data can be used to quantify phenotype adaptation. We write:

“The observation that cell count data alone can not be used to distinguish between semi-biased and biased phenotype adaptation opens up for a broader question: what type of experimental data is needed to identify phenotypic heterogeneity in cell populations and quantify how biased phenotypic adaptation is? We address this question in a subsequent study (Browning, 2024), where we have moved from a discrete to a continuous phenotype population model in order to allow for more rigorous mathematical analysis revealing that although cell count data alone is not sufficient to distinguish between the semi-biased and biased PU models, cell count data supplemented with information about birth and death events are. While we build our model on 3-day growth rates and (up to) four week treatment data, Russo et al. (Russo, 2022) combined long-term (over ten weeks) monitoring of experimental growth rates with an ordinary differential equation model to demonstrate that clinically approved targeted therapies induce drug-sensitive-to-tolerant adaptations in colorectal cancer cells. Further evidence of drug-induced cell plasticity was found by recent studies combining mathematical modelling and *in vitro* experiments (Gevertz2024, Wu2024). Beyond growth curve focussed studies, other methods to quantify phenotype adaptation have been developed for data sets in which individual cells can be experimentally categorised into phenotype groups by, e.g., fluorescent markers (Gupta, 2011; Buder, 2017). ”

Please see the Discussion section, line 272.

Note: Thank you for the suggested references which have all been included.

Point 10. On page 2, I'm not sure it's true that a phenotypic progression towards a drug-adapted state has not been applied in the interpretation of experimental data. One example is [10], where the phenotypic state is continuous, and another is the already cited [1], where the states are discrete. I think [7] also qualifies even though it involves single-stage evolution from drug-sensitive to drug-adapted. There is further work that has explicitly used a model with oneway transitions towards drug-resistance in the presence of drug and one-way transitions back in the absence of drug in the interpretation of experimental data, and they have found

a preference for intermittent/adaptive treatment over continuous treatment, for example [11, 12].

To address the reviewer's comment we have:

- Edited the text in the results to be more specific about what we mean. We now write: "This gamut of phenotype progression is highly studied in segments of the mathematical literature (Lorenzi, 2015) but, to the best of our knowledge, has not been applied in the interpretation of experimental data to select between candidate phenotype update models". Please see the Results section, line 130.
- Acknowledged/cited references 7 and 10, as outlined in our response 7iii below.
- Included a section on the adaptive therapy literature. We now write: "This ecological perspective has served as the theoretical underpinning of adaptive therapy, where treatment is designed to maintain a drug-sensitive cancer cell population that competitively excludes drug-resistant clones (Zhang, 2022; West, 2020). This existing framework has historically relied on an assumed cost-of-resistance, where drug-resistant clones are less fit than drug-sensitive clones in the absence of treatment. However, recent theoretical work has demonstrated that this assumption is not necessary for adaptive therapies to outperform conventional maximum-tolerated dose strategies (Strobl, 2022; Viossat, 2021). Here, in contrast with much of the adaptive therapy literature (Gatenby, 2009; Gatenby, 2020a), we use a simple model to capture improved treatment responses to intermittent therapy without explicitly modelling competition between clones." Please see the Discussion section, line 240.

Point 11. Pertaining to the discussion of mathematical modeling of cell plasticity and its impact on treatment responses and the evolution of therapy resistance on page 8, there is plenty of theoretical and computational work on this topic using simpler ODE and stochastic models, not just structured partial differential equations. See for example the references within [13].

To mention ODE models used for studying phenotypic plasticity we now write:

"Beyond melanoma, cell plasticity, and its impact on treatment responses and the evolution of therapy resistance, has been studied mathematically through ordinary differential equation models (Russo, 2022) and partial integro-differential equation models (Clairambault, 2019; Stace, 2020; Chisholm, 2016; Cassidy, 2021; Chisholm, 2015) which, notably, can be used to suggest model-informed treatment strategies that impede the evolution of resistance." Please see the the Discussion section line 302.

Please note that we cited references 7 (Russo, 2022) and 10 (Chisholm, 2015) as appropriate.

Point 12. Pertaining to the discussion in the second paragraph on page 8, epigenetic therapies targeting non-genetic modifications have been proposed and tested in clinical trials for a while, both alone and in combination with other drugs. It is worth acknowledging this and referring to some review articles.

We have briefly mentioned recent *in vitro* experimental work that identifies potential epigenetic targets in the context of BRAF inhibitor resistant melanoma response 6.

We further make the simplifying modelling assumption that cells can revert from state $x = 1$ to state $x = 0$ following a drug holiday. This is an approximation that is based on experimental results for the studied melanoma system (Kavran, 2022).

Assumption 1: Cells can revert from phenotype state $x = 1$ to phenotype state $x = 0$ and, by doing so, recover the growth rates of drug-naive cells.

[...] We note that *Assumption 1* may not be appropriate for all cell systems, as cell-level drug resistance in cancer ranges from reversible to irreversible (Bhat, 2024). *Please see Methods, line 369.*

Reviewer 1 - minor comments

Thank you for the feedback in minor comments 1-15. We appreciate that that the reviewer has pointed out where more/rephrased information would help readers and/or make the manuscript more precise. Thank you also for correcting some mistakes. We describe how we address these minor comments in points 1-15 below.

Point 13. I think “constituent activation” should be “constitutive activation”.

We have changed from constituent to “constitutive” activation. Please see Introduction line 17.

Point 14. I suggest describing what “epigenetic” means when it is first introduced.

We now explain the word epigenetic: “Such evolution is driven by epigenetic changes in cancer cells, i.e., changes in gene expression that do not involve changes in the cells’ DNA sequence (Easwaran, 2014).” Please see Introduction line 33.

Point 15. I suggest adding a brief explanation of the biological significance of phosphorylated ERK.

We now explicitly describe the effect of ERK on uncontrolled cell growth: “Phosphorylated, i.e., activated, ERK is often considered to be the main output of interest in pathway-level studies of BRAF-mutated melanoma. This follows from the fact that activated ERK phosphorylates multiple proliferation-promoting substrates in the nucleus and cytoplasm, which can drive uncontrolled cell growth and division (Wellbrock, 2010).” Please see Introduction line 36.

Point 16. The fact that upon drug removal, drug-adapted cells resumed growth rates “similar” to those of drug-naive cells, deserves further discussion. It’s an open question (and it may of course be context-dependent) whether drug-adapted cells can fully revert to the drug-naive condition, and the data in Figure 1(a) seems to indicate that drug-adapted cells proliferate slower off-drug than drug-naive cells.

In the newly added methods subsection "Designing the fitness matrix", we clarify that the growth rates of drug-naive cells and cells that have recovered after a drug holiday are *similar*, and that we make the simplifying modelling assumption that they are the same.

Point 17. It would have helped me if it was said more clearly that the biased models apply both on and off drug, and that the move towards a higher-fitness phenotype is in different directions depending on the dose.

We have added this clarification in the results section: "In the implementation, the same update rule applies both on and off treatments, but the directionality of the updates in the semi-biased and biased phenotype update strategies is drug dose dependent." Please see Results, line 138.

We have also added a clarification in the methods section: "In biased update models, the directionality of the phenotype adaptation is dependent on the applied drug dose, as determined by the data. As such, cells move towards states of higher proliferative net fitness (Figure 2a,c). To fit the free parameters, we further assume that the same phenotype update strategy applies both on and off drugs.

Assumption 3: The same phenotype update strategy applies both on and off drugs.
Please see Methods line 416.

Point 18. 6. P2: On first read, I was confused about how the "resolve cell division and death" + "update phenotype states" steps in Figure 2(b) were implemented. Upon second reading, I understand from the second paragraph on page 2 that you either implement a pure-birth or pure-death model. Furthermore, the phenotypic updating seems to happen independently of cell division events, which I think is reasonable. However, I suggest laying out more clearly how the model is implemented in the main text, and also adding to the methods section how the model is simulated in algorithmic step-by-step form. This information is currently a little bit scattered and it was hard for me to get the full picture on a first read.

To make the overall modelling procedure and implementation more clear, we have expanded the methods sections. We have included a new methods subsection called **Implementing cell division, cell death and phenotype updates**. This subsection includes a description of the implementation (Figure 2b) in step-by-step algorithm form.

Please see the new subsection and Algorithm in Methods, line 439.

Point 19. In the fourth paragraph, I think "no update and biased" within the parentheses should be "no update and unbiased".

We have corrected the text to say "(no update and unbiased)". Please see Introduction line 123.

Point 20. In the fifth paragraph, ρ appears without having been introduced in the text.

We have edited the text to make the introduction of ρ more clear. We now write:

“Simulation results pertaining to 0 and 300 nM treatment responses have been produced with dose-dependent net growth rates from the fitness matrix (Figure 1d) and ρ , which denotes the probability that a cell changes its phenotype state at a programmed update time, set to $\rho = 1$ (Figure 2a).” *Please see Results, line 143.*

Point 21. At the end of the first paragraph, it would help to refer to the Methods section for a more detailed description of the parameter fitting.

In the result section, we now refer to the methods section for details on the parameter fitting. We write: “...The number of cellular phenotype updates per day, η_{on} and η_{off} are also chosen to best match the data (details about the parameter fitting are available in the Methods section).” *Please see line 160.*

Point 22. In the parameter fitting, I understand why the growth rate of drug-adapted cells is fit to the data, but I don't understand why the growth rate of drug-naive cells under 500 nM is allowed to be free. It seems most reasonable to use the information available from the multiple-dose experiment in Figure 1(a) and do a log-linear interpolation, especially since the data seems to be limited. Is there a reason for this choice?

To better motivate why we did not use log-linear interpolation in Figure 1d, we now write that:

As is shown in the Supplementary Material (Figures S1-S2), fitting η_{on} and η_{off} only, while obtaining FM(0,500 nM) and FM(1, 500 nM) from log-linear interpolation between 300 nM and 1000 nM growth rates, does not result in simultaneously data-matched simulations for continuous and intermittent treatments.

Please see line 434 and Figures S1-S2 in the newly added Supplementary Material.

Point 23. I was confused when looking at Figure 2(a) for the first time. It would probably help to explain how it depends on the drug dose whether the top or bottom row applies.

We have clarified the Figure 2a caption by adding this sentence: “The directionality of the net growth rates, i.e., whether the top or bottom row applies, depends on the applied drug dose and is inferred from growth rate data.” *Please see Figure 2.*

Point 24. In Figure 2(a), for the semi-biased update case, would it not be more logical to use ρ and $1-\rho$ for the probabilities, with the understanding that ρ is a number between 0 and 1?

We have clarified in Minor Remark 8 our definition of ρ , and hope that explains why we use $\rho/2$ in Figure 2a. This choice makes the meaning of ρ programatically consistent between the four regarded phenotype update strategies.

Point 25. In the caption of Figure 2(b), it is said that “each simulated day, cells update their phenotype states before/after the resolution of cell division and death with probabilities 0.5/0.5”. Is 0.5/0.5 here referring to the probability that the updates happen before or after the resolution of cell division and death? Are phenotypic updates not also allowed outside cell divisions, if they happen more frequently than once per day? Following my earlier comments, it would help to describe the implementation of the model more clearly in the main text, methods and Figure 2.

To address these questions, we have edited the Figure 2b caption to: “Each simulated day, cells update their phenotype states η times either before (with probability 0.5) or after (with probability 0.5) the resolution of cell division and death.” We have also added an “ η times” in the Figure 2b flowchart. *Please see Figure 2b.*

Please see also our response to minor comment 6 regarding our, now more detailed, implementation section in Methods.

Point 26. In Figure 2(c), it would be more logical to me if day -28 showed 10 small columns each of height 0.1, since presumably the idea is that each column represents the proportion each phenotype makes of the entire population. It also seems like the y-axis label should refer to proportion as opposed to count.

We have edited Figure 2c to show phenotype densities (so that the total sum of the bars, and the height of the vertical axis, is one in all plots). The figure caption has, accordingly, been updated to say “*densities*” instead of “cell counts”. *Please see Figure 2c.*

Point 27. 15. P6: Are the $\eta = 1$ and $\eta = 4$ cases mislabeled in Figure 3(c)?

The variable η (i.e., $\eta = 1, \eta = 4$) was indeed mislabeled in Figure 3c,e, thank you. We have now corrected this. *Please see Figure 3.*

Response to reviewer 2

Overview and Major Claims: This well written and clear paper presents a mathematical model of melanoma adaptation and the authors use it to explore response to treatment. This work focuses specifically on mutations in BRAFV600E and treatment with the BRAF-inhibitor encorafenib. The paper focuses on multi-state phenotypic model with various methods of phenotype updating. The authors compare different undirected (none, unbiased) and directed (semi-biased, biased) modes of phenotypic evolution. These models of resistance simulate phenotypic plasticity (unbiased, semi-biased, biased updating) in addition to resistance arising from pre-existing heterogeneity (no updating). The authors present the first use of this model type for the interpretation of experimental data. The authors show that in their model, directed adaptation is required to reproduce experimental observations. Additionally, they claim that it provides a growth rate data driven explanation for why intermittent treatments outperform continuous treatments at 500 nM encorafenib. They go on to explore the extent to which intermittent treatment regimes outperform continuous maximal dosing for the directed phenotypic update model of resistance. The discussion, methodology, and results are clear, with minor updates needed for clarity and context. While the model is somewhat agnostic to biological mechanism, the authors discuss melanoma specific biology to appropriately contextualise their findings.

Novelty: The novelty of this work lies in the combination of phenotypic state modeling with experimental data in melanoma. The authors also present novel work in the further use of their fitted model in making predictions under clinically relevant treatment schedules and treatment holidays. They present interesting findings around the non-monotonic relationships between adaptive behavior and treatment scheduling.

Convincing: Yes, except Fig 3c.

Validity of Statistical Approach: Mathematical model appropriately constructed and defined.

Ease of Reproduction: Appropriate level of detail included for figures. Code provided (github).

We are grateful to Dr Rowan Barker-Clarke for the corrections and feedback on how to improve the scientific content and presentation of our manuscript. We understand Dr. Barker-Clarke's concerns and describe how we address them in points 1-5 below.

Concerns

Point 28. Concerns (Fig 3.) This figure is a little difficult to parse, perhaps due to missing visual schema on dosing and empty boxes and because initial condition subfigures seem to be a little excessive/add clutter. Green dashed lines are under-explained. In c) and f) It is unclear why there are transparent boxes for 0nM and 500nM encorafenib. The legend does not mention 0nM. Perhaps there is meant to be a colored intermittent dosing regime not shown. Importantly, in Fig. 3c) The n=4 updates (dashed) per day has a lower cell count than n=1 (solid), this seems to oppose to the claim in the text (that treatment is more effective when updates per day are reduced).

We have edited Figure 3, to make it easier to parse. Specifically:

- In panels c,d we have made the “drug on” boxes a darker gray, and edited the legends to more clearly communicate the treatment schedule, i.e., when the drug is on (500 nM) and when it is off (0 nM). We have removed the stars to simplify the

figure and remove clutter. We have also corrected the η -labeling ($\eta = 1, \eta = 4$) so that, in consistency with the text, the treatment is more effective when the updates per day are reduced from 4 to 1 (we had mislabeled the η s in the original manuscript figure, thank you for drawing attention to this).

- For panels b,e we now describe the green dashed line in more detail in the figure caption. Specifically, we write: **Based on mean values from 100 simulations, heatmap bins show effective growth rates in response to different phenotype updates per day (η), and on/off treatment intervals (T). Effective growth rates for 300 nM (left) and 500 nM (right) continuous treatments are highlighted in the colour legends. In the 300 nM heatmap, all shown $\eta - T$ pairings yield lower effective growth rates than the continuous treatment. In the 500 nM heatmap, the green, dashed curve separates the $\eta - T$ pairings into two groups: pairings that yield lower effective growth rates than continuous treatments, and pairings that do not. The curve is obtained through mathematical analysis (see Methods).**
- Note that we have chosen to keep panels a,d to clearly communicate the difference between the top (a-c) and bottom (d-f) part of the figure. However, we hope that the aforementioned edits resolve these issues with the figure.

Please see Figure 3.

Point 29. Fig 1. d) There is a mix of significant figures and decimal places, consistency (and increased text size) would be beneficial in the presentation of this subfigure. Bolded yellow on gray is also difficult to read and the authors may wish to choose something like underlining the negative growth rates instead.

We have improved Figure 1d (the fitness matrix). Specifically we now have:

- Three digits after the decimal points in all matrix elements. We have also updated the inferred growth rates in the Methods section, line 351 updated accordingly.
- Negative net growth rate values as underlined instead of yellow-fonted. We have updated the figure caption accordingly.
- Increased font size in the matrix elements.

Please see Figure 1d.

Point 30. Fig 3. b) The legend and main text would benefit from rephrasing and explanation about the analytical result, and at least a direction to the analytical results which have been placed in the methods. The authors may wish to highlight the lack of a boundary at 300nM.

To clarify the analytical result we have:

- Edited the Figure 3 caption to better describe the analytical result curve (please see the second bullet point in our first response item to this reviewer).
- Pointed to the methods section. We now write: "Using the biased update strategy, we derive analytical approximations of effective net growth rates λ_{eff} obtained in the limit of large total treatment times (see the Methods section for details). " *Please see line 193.*

- Added an explanation in the Results section on the lack of a green curve in the 300 nM heatmaps. We now write: “These pairings are highlighted by yellow regions in Figure 3b,e and are separated, by analytically derived curves, from $\eta - T$ pairings that result in higher effective growth rates than continuous treatments. Note that in the 300 nM heatmaps, all visualised $\eta - T$ pairings outperform the continuous treatment strategy, hence no separating curve is shown.” Please see line 199.

Point 31. The authors use very specific parameters including numbers of states and assumptions (linear interpolation between states, uniform seeding). While these assumptions are often discussed, emphasis on the choice (literature, fitted optimal or arbitrary) of key parameters or parameter ranges when they arise would improve manuscript clarity. (Primarily because fitted parameters are Results themselves).

In response to these concerns, we have restructured and extended the Methods section to better explain how we motivate our modelling choices and assumptions. See the highlighted (red) text in the Methods section, particularly the following assumption and associated discussions:

- *Assumption 1: Cells can revert from phenotype state $x = 1$ to phenotype state $x = 0$ and, by doing so, recover the growth rates of drug-naive cells.*
- *Assumption 2: Cells move between phenotype states $x = 0$ to $x = 1$, and back, in $n + 1$ discrete steps.*
- *Assumption 3: The same phenotype update strategy applies both on and off drugs.*

Please see lines 373, 384, 422 for the three assumptions, respectively.

We have also included a supplementary figure (S3) demonstrating simulation results for other choices of phenotype granularity (i.e., n). In the main manuscript we write:

“The Supplementary Material (Figure S3) also includes simulated cell counts in response treatments for different choices of phenotype granularity, i.e., n , showing that if we consider the cell population to be phenotypically non-binary, then the directed strategies better explain the observed data than the non-directed strategies.” Please see line 164.

Moreover, to make our work more generalisable (and less focussed on specific parameters), we have included a Wolfram Mathematica notebook that lets users derive effective growth rates for any choice of treatment on and treatment off durations. The notebook is included in **Supplementary Material SM2** in printed format, and on the GitHub page in notebook format.

Point 32. The authors neglect to mention (even in discussion) how their explanation for intermittent therapy efficacy fits into the literature that explains intermittent schedule success via ecological/frequency dependent effects. Specifically so called “adaptive therapy” can provide alternate data-driven explanations for the success of intermittent therapies (see adaptive therapy literature and models).

We now discuss our work in relation to adaptive therapy. We write:

“One notable consequence of our results is that dynamic phenotypic adaptation alone suffices to explain why intermittent treatments can outperform continuous treatments.

This explanation offers an alternative to variations of the more common inter-clonal competition-models in which two or more distinct subpopulations of cells, such as a drug-resistant and drug-sensitive clones, interact with each other and compete for resources (Madan, 202). This ecological perspective has served as the theoretical underpinning of *adaptive therapy*, where treatment is designed to maintain a drug-sensitive cancer cell population that competitively excludes drug-resistant clones (Zhang, 2022; West, 2020). This existing framework has historically relied on an assumed cost-of-resistance, where drug-resistant clones are less fit than drug-sensitive clones in the absence of treatment. However, recent theoretical work has demonstrated that this assumption is not necessary for adaptive therapies to outperform conventional maximum-tolerated dose strategies (Strobl, 2022; Viosat, 2021). Here, in contrast with much of the adaptive therapy literature (Gatenby, 2009; Gatenby, 2020a), we use a simple model to capture improved treatment responses to intermittent therapy without explicitly modelling competition between clones.”

Please see line 240.

Response to reviewer 3

The paper, "Growth rate-driven modelling reveals how phenotypic adaptation drives drug resistance in BRAFV600E-mutant melanoma," provides a framework for understanding the role of phenotypic adaptation in cancer drug resistance. Through a computational model informed by experimental data, the authors explore how melanoma cells adapt their phenotypes in response to BRAF inhibitor treatments. Their model incorporates discrete, plastic phenotype states that dynamically respond to drug exposure, revealing that adaptation is directed towards high-fitness states. This mechanism explains the observed efficacy of intermittent treatment schedules over continuous dosing, offering a new perspective on optimizing therapy strategies. The integration of growth rate data and simulation experiments showcases the model's robustness and potential translational value.

In general, I like the paper and would like to ask the authors to provide some explanation of how they generate the Fitness Matrix.

Thank you for reviewing our manuscript and prompting us to more clearly describe how we designed the fitness matrix.

Point 33. In response to your feedback, we have now included a new subsection in the methods section called "*Designing the fitness matrix*". *Please see line 357.*

Response to Reviewer 1

I appreciate the authors' efforts to address my comments. Most of my comments have been addressed well, but in my opinion, the most substantial concerns raised in the first three comments of my previous report have not been addressed. Based on the authors' responses, I'm not sure they can be addressed other than to more clearly acknowledge the limitations of the study. More specifically:

Overall response: Thank you for the continued feedback. We have revised the manuscript to address all of your remarks as is detailed in the point-by-point responses below. Please note that new and edited text in the manuscript is highlighted.

Point 1. I mentioned in my previous report that it's reasonable to assume that cells incubated in doses higher than 500 nM will adapt to tolerate those doses and the authors seem to agree with this. The same logic applies to doses below 500 nM, in that cells incubated in 100 nM for example may adapt to this dose, so that they have increasing growth rates up to that dose and decreasing growth rates at higher doses. In addition, the data does not include drug rechallenge at 500 nM, which makes it difficult to infer what the growth rate is even at 500 nM. The authors describe $F_M(x, d)$ as giving the growth rate of phenotype state x at dose d (stated directly in lines 360–362 and implied elsewhere), and it would be reasonable for the reader to assume that the fitness matrix can be used to make (good) predictions about the outcome of continuous and pulsed treatments involving all doses, whereas I believe the data is not sufficient to derive a matrix for this purpose. If the fitness matrix is included, I think it should be made clear that it is valid for describing state-dependent growth rates for 500 nM treated cells rechallenged with the drug at the indicated doses, and that it can possibly be interpreted more broadly under the seemingly strong assumption (unless there is some evidence that this assumption is valid) that cells incubated at other doses will adapt specifically to the 500 nM dose. It would also be useful to discuss (in the main text) what kind of experiments would be necessary to derive a fitness matrix in the way the authors have conceived it. In my opinion, this is a significant issue, and it's better to clearly acknowledge and discuss the limitations of the data than to make unwarranted inferences.

To address the concerns raised by the reviewer in this point, we have edited the manuscript to better explain the limitations and scope of our study. Details are provided in the bullet points below.

- **In the introduction we** now address shortcoming of the inferred fitness matrix and write (line 105): *Note that the fitness matrix approximates net growth rates from the three-day drug-challenge assay for drug-naive cells ($x = 0$), cells pre-incubated at 500 nM encorafenib ($x = 1$), and intermediate phenotype states $x \in (0, 1)$. In Figure 1d, the yellow line separates doses above and below the pre-incubation dose of 500 nM, highlighting that drug-resistant phenotypes may continue to increase their resistance beyond the three-day assay period at doses above 500 nM. Likewise, cells exposed to doses below 500 nM encorafenib may continue to adapt their net growth rates after the three first days. The absence of longer-term growth rate measurements is a limitation of our study, and in the Discussion we outline how future experiments could be designed to address this and yield more informative fitness matrices.*

- **We have edited Figure 1d** to include the above mentioned yellow line. The caption now includes the description: *The yellow line separates doses above and below the pre-incubation dose of 500 nM.*

- **We have edited the discussion-text** to more clearly describe the direction of future experiments to map out more detailed fitness matrices (**line 284**): *“The model developed in this study was intentionally kept simple to enable direct mapping between phenotype states and growth rates. While this simplicity facilitates interpretability and has yielded informative results, it also highlights directions for more sophisticated model–experiment integration that could provide deeper quantitative insight into phenotypic adaptation. Future experimental designs could explicitly separate measurements of cell division and death events, enabling models to incorporate growth and death rates independently rather than relying on net growth rates alone. Further, our current approach of linearly interpolating between drug-naive and drug-adapted growth rates to assign fitness to intermediate phenotype states could be refined by systematic drug-challenge assays. Specifically, measuring cell counts at multiple time points after pre-assay incubation across a range of drug doses would allow the mapping of growth rates in the phenotype–dose parameter space. It would also be informative to test whether, and under which conditions, drug-adapted cells can fully regain the growth rates of drug-naive cells. Such data would permit estimation of dose-dependent adaptation velocities and directly inform ongoing debates on whether cell-level drug resistance is reversible in different cancer types (Bhat, 2024). Incorporating sequencing alongside these assays would further aid in evaluating underlying molecular changes, thereby simultaneously linking phenotypic adaptation to both growth rates and genetic/transcriptomic mechanisms.”*

- **We now more often remind the reader of the study scope** by more explicitly writing:

- **Abstract (line 13)**: *“The model subsequently provides an explanation for when and why intermittent treatments outperform continuous treatments in the studied system...”*
- **Introduction (line 60)**: *“Subsequent comparisons of model-generated cell populations and in vitro cell cont data reveal that phenotypic adaptation is directed towards phenotype states with high net growth rates in the regarded cell system.”*
- **Discussion (line 302)**: *“Whilst our results clearly indicate that phenotypic adaptation is directed towards state of high fitness in the regarded in vitro experiment,...”*

Point 2. In the second comment of my previous report, my concerns were aimed at Figure 3, whereas in their response the authors only address Figure 2. I agree that the distinction between 300 nM and 500 nM is not important for Figure 2. However, in Figure 3 the results are shown as being significantly different for 300 nM and 500 nM, which is not based on model fitting but on a log-linear interpolation between the experimental observations under 300 and 1000 nM. The authors have acknowledged that they cannot say with great confidence that the dynamics differ between these two doses. Therefore, in my opinion, Figure 3 should not display results that indicate a significant difference, especially given contradictory results from the model fitting in Section 2. I think it should also be acknowledged in the text that the log-linear interpolation may not accurately reflect the dynamics at 500 nM, since this assumes that the growth rate of drug-adapted cells decreases from 300 to 500 nM, whereas in my opinion it is reasonable to assume that it will either remain the same or increase from 300 nM to 500 nM.

To address your comments in the main text we now explicitly say that the visual difference between 300 and 500 nM follows from the log-linear interpolation, and that this

interpolation is a simplifying modelling assumption (bullet point 1 below). We also highlight that the colour scales differ between heatmaps (bullet point 2). Lastly, we have improved the motivation of the study-objective related to Figure 3, focussing on the value of the Figure 3-related method development and mathematical modelling (that have value independently of the quantitative choice of log-lin inferred net growth rates at 500 nM). Details below.

- **We comment on the differences of the heatmaps on line 226:** *‘Note further that throughout Figure 3, we have used phenotype-dependent net growth rates for 500 nM that are obtained through log-linear interpolation between net growth rates for 300 and 1000 nM in the fitness matrix (Figure 3a). This interpolation is used due to three-day growth rates in 500 nM being missing in the data, and likely does not capture the full drug-adaptation of the cells which contributes to the visual differences between the 300 and 500 nM heatmaps in Figure 3.’*
- **We comment on the heatmap color scales in the Figure 3 caption:** *‘Note that the colour scales differ between heatmaps to clearly illustrate the impact of η and T for a fixed dose and initial condition.’*
- **We motivate the Figure 3-related results beyond the quantitative choices made (line 254):** *‘Although the quantitative results presented in Figure 3 are problem-specific, the qualitative demonstration that treatment scheduling together with phenotypic adaptation shape optimal strategies has broader relevance beyond this study and motivates treatment designs that includes monitoring if and how phenotypic expression changes during treatment.’*

Point 3. The question stated at the beginning of the third results section is: “Given directed phenotypic adaptation, when do intermittent treatments outperform their continuous counterparts?” This is a quite general statement, but the authors proceed to investigate how a 500 nM 1:1 intermittent treatment compares to 500 nM continuous treatment, depending on the number of phenotype updates per day (η) and the interval length in days (T). How does this investigation relate to the system and the data being studied in the paper? Is the idea that different systems will have different values of η and that this approach can identify for which systems 500 nM continuous treatment is better than 500 nM intermittent treatment? If so, the problem is that the model fitting procedure used to infer η relies on data that already shows whether an intermittent treatment is better than continuous.

- We do study the posed question broadly (beyond the regarded melanoma study-system) through the formulation of the ODE modelling methodology presented in Eqs. 4-7. Although we (of course) have to put in some system-specific model parameters to generate simulation results in Figure 3, the ODE methodology can be analysed symbolically (see Eq. 7 and S3 in the Supplementary Material) and is modifiable to study other systems.

We now clarify this in the results section (line 211): *‘By comparing the λ_{eff} values resulting from continuous and intermittent treatments, we can evaluate which treatment is more effective at keeping λ_{eff} low, and thus impeding cell population growth. While we, in Figure 3, focus on intermittent treatments with 1:1 scheduling such that $t_{\text{on}}=t_{\text{off}}=T$, the developed ODE formulation and methodology can be adapted to study systems with arbitrary schedules and net growth rates.’*

And the methods section (line 557): *‘We remark that Eq. 4 can be modified to investigate scenarios where phenotype adaptation occurs at different rates during drug-on and drug-off periods, by allowing the drug-off rate to differ from the negative of the drug-on rate (further, these rates need not be constants). Similarly, Eq. 6 can be modified to model intermittent treatment schedules with on/off proportions other than 1:1, by altering the integration limits. Thus, the ODE framework presented in Eqs. 4-6 can be readily adapted to study a broad range of phenotype adaptation and treatment settings. For such modifications, deriving new effective growth rates λ_{eff} is straightforward using the procedure outlined in the Supplementary Material (S3).’*

- In Figure 3, the purpose of letting η and T vary across the heatmap-axes is to demonstrate (mathematically/computationally) that these two model parameters (η and T) affect treatment responses. This is biologically relevant as T is something that humans can control, and phenotypic adaptation (thus η) has been proposed as a treatment target. We have edited the first paragraph in the third result section to (i) better motivate the purpose of the simulations shown in Figure 3, and (ii) be more clear about what we mean when we say that we address the self-posed question (**lines 199**): *“Our results raise the question: Given directed phenotypic adaptation, when do intermittent treatments outperform their continuous counterparts? As a step to address this question computationally and mathematically, we perform a series of simulation experiments in which output dynamics are quantified in response to variations of two model-inputs: the number of times per day that a cell can update its phenotype state (η), and the duration of treatment on/off intervals ($t_{\text{on}}/t_{\text{off}}$). Varying these parameters allows us to investigate the impact that they have on simulated treatment responses.”*

Point 4. In general, I’m not convinced a lot of value is being added by fitting the mathematical model to the data, in terms of specific novel biological or therapeutic insights. The authors show that the preference for intermittent treatment under 500 nM can be explained by a model with one-way transitions to a drug-adaptive state on drug and one-way transitions to sensitivity off-drug. However, it’s not surprising that this is a potential explanation (as pointed out previously, other authors have used this idea to obtain a preference for intermittent/adaptive therapies, see for example [1, 2]), and doing anything with this information presumably requires a biological investigation into the mechanism of drug adaptation. Once the model has been fitted to the data, as far as I can tell, nothing more can be said about treatment on that system with that drug, in particular no inferences can be made about intermittent or continuous strategies with doses other than the one already tested. If this was possible, fitting the model would enable making testable predictions beyond the experimental result and would help in discovering better dosing strategies. The authors do contemplate how slowing down the rate of adaptation can improve treatment outcomes, but this investigation is quite generic and implementing this idea presumably requires discovering the mechanism of drug adaptation, identifying an agent capable of inhibiting the mechanism, and understanding the dose response of the agent, so I’m not sure the model is adding significant value there either.

We emphasise on two main contributions of our study below:

- **Main contribution 1:** Many mathematical modelling studies use one of the phenotype update models illustrated in Figure 2a. In our response to point 6 below, we further discuss studies that use the unbiased update rule. Our study systematically evaluates all four update models against each other, based on their ability to reproduce experimental growth curves in response to treatments (Figure 2c), and finds that directed

phenotype adaptation is more likely than undirected adaptation in the studied cell system. The framework derived in our work can therefore be applied by mathematical modellers to assess which phenotype update models are most plausible in their regarded systems, or alternatively, our main conclusion –that phenotype adaptation is directed in the melanoma system studied in our study– can directly inform model design when applicable.

- **Main contribution 2:** We present a mathematical ODE framework for evaluating continuous treatment responses against intermittent ones. The equations derived within this framework are applicable to a wide range of phenotype adaptation and treatment settings, as further elaborated in the final bullet point of our response to point 6 below.

To make these two main contributions more clear in the manuscript we have:

- Edited the title to: “*Growth rate-driven modelling suggests that phenotypic adaptation drives drug resistance in BRAFV600E-mutant melanoma*”.

- Highlighted the contributions in the abstract (**line 8**): “*To describe how cells transition between phenotype states, we explore a gamut of candidate models common in the mathematical biology literature. Comparing these on their ability to reproduce in vitro growth curves, data-matched simulations suggest that phenotypic adaptation is directed towards states of high net growth rates, enabling the evasion of drug-effects. The model subsequently provides an explanation for when and why intermittent treatments outperform continuous treatments in the studied system, and demonstrates the benefits of not only targeting, but also leveraging, phenotypic adaptation in treatment protocols. Building on the IBM, we present a flexible mathematical methodology based on ordinary differential equations to compare responses to continuous and intermittent treatments through long-term effective net growth rates.*”

Point 5. In lines 60–62, the authors have reworded the description of their framework, saying that the framework “provides a mechanistic explanation for when and why intermittent treatments may be more effective than continuous treatments”. I don’t agree that a mechanistic explanation is being provided in the way I understand this term, since the biological mechanism behind drug adaptation is not being modeled.

We acknowledge the ambiguity in the term mechanistic, as it can be applied at different biological scales (or levels). In our work, mechanistic refers specifically to the level of phenotype classification, rather than to molecular-level modelling of interactions between cellular subcomponents and drugs.

To clarify this, we now write (**lines 65**): “*Based on multiple-dose growth rate data from untreated and treated cells, our framework thus provides a mechanistic explanation (on the phenotype classification level) for when and why intermittent treatments may be more effective than continuous treatments.*”

Point 6. If I understand the discussion in lines 124–128 correctly, it seems to be based on the premise that phenotype instability is synonymous with an unbiased update model. I’m not sure I agree with that premise and don’t see why unbiased updates are a reasonable null model. Transitions between states are often due to epigenetic mechanisms like methylation, and it is known that the gain and loss of methylation are driven by distinct biological mechanisms. So why would we expect transitions between cell states to be unbiased in the absence

of stress? There is also experimental evidence that transition rates between states are in general distinct in the absence of stress/drug, see for example the previously referenced [3]. This includes examples where some cell types are more resistant than others, see for example [4].

- We did not mean to infer that the unbiased model is *fully synonymous* to phenotypic instability (but thank you for pointing out that it read as such). Rather, we were suggesting a possible biological interpretation, referencing previous work by Crisholm (2015). To clarify this, we now write **(line 135)**: “*Biologically, the unbiased PU strategy could correspond to phenotypic instability, while the biased PU strategy corresponds to stress-induced phenotypic adaptation, in accordance with work by Chisholm et al. (Chisholm, 2015).*”
- We included the unbiased model as a null model largely because it frequently appears in the mathematical modeling literature. For instance, it is treated as the sole mode of phenotypic adaptation in (Lorenzi, 2025), and competition-style ODE models that include spontaneous phenotype switching are not uncommon. Furthermore, in the referee’s reference [5], cells are allowed to switch spontaneously within groups of states labeled ‘sensitive’ and ‘resistant’. Comparing [5] to our model, we do not define resistant and sensitive groups of phenotype state; instead, drug sensitivity is represented along a gradual grid, allowing transitions between all states. However, the point that we want to make here is that unbiased phenotype update models *are* commonly considered in the modelling community. Therefore we argue that it is not only reasonable, but also informative, to include the unbiased candidate model in our study.
- We are not arguing for that the unbiased model is the most biologically plausible phenotype update model (but it is, as argued above, a commonly used one). Actually, one of the main results in our paper is that the unbiased model is less likely than directed phenotype models to explain the cell population dynamics seen in Figure 2c for 500 nM.
- We do allow switching rates to be different when the drug is on or off when we fit the model parameters to data through simulations (hence the two model parameters $\eta_{\text{on}}, \eta_{\text{off}}$ that appear in Eq. 3).
 - We now clarify this in Assumption 3 **(line 495)**: “*Assumption 3: The same phenotype update strategy applies both on and off drugs, and the update rates may differ between drug-on and drug-off days.*”
 - We now cite the referees reference [3] to even more motivate that different switching rates are allowed **(line 493)**: “*To fit the free parameters, we further assume that the same phenotype update strategy applies both on and off drugs, although the phenotype update frequency may vary between drug-on and drug-off days*[3].”
- In addition to the above bullet point, the mathematical analysis presented in this paper also allows for different switching rates. Specifically, in the ODE model presented in Eqs. 4-7 and shown in the green-dashed result curves in Figure 3b,e, the ‘drug on’ and ‘drug off’ phenotype adaptation rates (now $\omega, -\omega$, respectively) can easily be modified in Eq. 4 to be asymmetric. Of course then the analytical results in Eq. 7 would not hold as these are derived for the (on,off) rate-pair $(\omega, -\omega)$, but it would be straight-forward to derive updated effective growth rates for asymmetric adaptation rates. This is now highlighted in the text **(line 557 – see previous text insert in Point 3; Paragraph 3)**.

Reference

Lorenzi T, Macfarlane FR, Painter KJ. Derivation and travelling wave analysis of phenotype-structured haptotaxis models of cancer invasion. *European Journal of Applied Mathematics*. 2025;36(2):231-263. doi:10.1017/S0956792524000056

Point 7. In line 304, when mentioning mathematical study of cell plasticity through ordinary differential equation models, I think it's more appropriate to reference [5] as suggested in my previous report, which is more of a review and covers both ODE and stochastic models. There are other reviews/perspectives like [6, 7] which are also appropriate to reference to indicate that the amount of work on this topic.

We now include references [5,6,7]. Since they are important, but do not explicitly focus on one type of mathematical modelling method, we cite them in the following way on line 306: *"Beyond melanoma, cell plasticity, and its impact on treatment responses and the evolution of therapy resistance, has been studied mathematically [5,6,7] through ordinary differential equation models..."*

[1] E. Kim, J. S. Brown, Z. Eroglu, and A. R. Anderson, "Adaptive therapy for metastatic melanoma: predictions from patient calibrated mathematical models," *Cancers*, vol. 13, no. 4, p. 823, 2021.

[2] A. Yin, J. G. van Hasselt, H.-J. Guchelaar, L. E. Friberg, and D. J. A. Moes, "Anti- cancer treatment schedule optimization based on tumor dynamics modelling incorporating evolving resistance," *Scientific Reports*, vol. 12, no. 1, p. 4206, 2022.

[3] T. Buder, A. Deutsch, M. Seifert, and A. Voss-Böhme, "Celltrans: an r package to quantify stochastic cell state transitions," *Bioinformatics and biology insights*, vol. 11, p. 1177932217712241, 2017.

[4] P. B. Gupta, C. M. Fillmore, G. Jiang, S. D. Shapira, K. Tao, C. Kuperwasser, and E. S. Lander, "Stochastic state transitions give rise to phenotypic equilibrium in populations of cancer cells," *Cell*, vol. 146, no. 4, pp. 633–644, 2011.

[5] P. Jain, A. S. Duddu, and M. K. Jolly, "Stochastic population dynamics of cancer stemness and adaptive response to therapies," *Essays in Biochemistry*, vol. 66, no. 4, pp. 387–398, 2022.

[6] J. Foo, D. Basanta, R. C. Rockne, C. Strelez, C. Shah, K. Ghaffarian, S. M. Mumenthaler, K. Mitchell, J. D. Lathia, D. Frankhouser, et al., "Roadmap on plasticity and epigenetics in cancer," *Physical biology*, vol. 19, no. 3, p. 031501, 2022.

[7] F. J. Whiting, J. Househam, A.-M. Baker, A. Sottoriva, and T. A. Graham, "Phenotypic noise and plasticity in cancer evolution," *Trends in Cell Biology*, vol. 34, no. 6, pp. 451–464, 2024.

Response to Reviewer 2

Thank you for the helpful feedback in this and previous revisions. We have corrected the pointed out spelling error.

~~Response to Reviewer 3~~

No reviewer 3 in this round of revisions.

Response to Reviewer 4

This manuscript uses a simple theoretical model to understand the role of phenotype plasticity in drug resistance in cancer. Specifically, the authors formulate a computational individual cell model (also called an agent-based model), where each cancer cell can either divide or die based on data-driven net growth rates in BRAFV600-mutant melanoma cells adapted from a paper by Kavran et al. in PNAS. The rates are a function of a phenotype state x_i assigned to each cell, and are derived from the growth rate data, where cells are either pre-exposed ($x_i = 1$) or naive ($x_i = 0$) to pre-treatment by the BRAF-inhibitor encorafenib at a dose of 500 nM. As opposed to two states, the authors consider n intermediate phenotypic states and corresponding growth rates, which are obtained via log-linear interpolation between $x_i = 0$ and $x_i = 1$ ($n = 9$ in the main body of the work). They then formulate four working hypothesis regarding the relative bias of this adaptation process, which is assumed to occur during treatment. The relative biasing of this process is then used to describe the degree to which the drug is inducing resistance. Using an additional data set from the same work by Kavran, the authors consider the question of treatment holidays, which were found to be experimentally superior compared to continuous therapy. Fitting each of the four hypotheses to this data, the authors find that only two of the four scenarios are able to capture the observed increase in response to intermittent therapy, which thus serves as a form of model selection, as these two hypothesis involve drug induced resistance (semi-biased and biased updates). Although they cannot distinguish between these two scenarios, their results show evidence for some degree of drug-induced resistance in the presence of encorafenib in the WM239A cell-line studied. Overall, I believe the article is very interesting, and is worthy of publication in Communications Biology. I particularly enjoy the simplicity of the model, and its connection to experimental data. It is also very well-written, with most conclusions very clearly demonstrated from the proposed methods. I do have a few comments, listed below, which I believe will make the article stronger. Once these are addressed, I believe the article should be accepted for publication.

Overall response: Thank you for the constructive feedback on both scientific content and presentation. We have addressed all of your remarks and questions, as is detail below. Please note that new and edited text in the manuscript is highlighted.

Reviewer 4 - main comments

Point 1. My major concern is the lack of explanation at the reason why intermittent therapy outperforms continuous therapy in the cases of the semi-biased and biased update strategies. I believe (but please correct me if I am wrong) is that this is due to the fact that $x = 0$ is more fit at dose = 0 nM, and $x = 1$ is more fit at dose = 500 nM, i.e. that there exists a switch at in the dose-response curve between doses 10 and 30 nM in Figure 1 (b). Indeed, it is intuitive, and I am sure could be demonstrated with an even simpler theoretical model: when the drug is on, cells transition (in the semi-biased and biased cases) from $x = 0$ to $x = 1$, producing a resistant population in mostly state $x = 1$. However, when treatment is stopped, the population goes back from $x = 1$ to $x = 0$, since $x = 0$ is more “fit” in this environment. When treatment is again resumed, the population is mostly sensitive ($x = 0$), and thus therapy is more again effective. Note that continuous therapy does not provide this “chasing” behavior in those biased cases, and hence won’t be as effective. I believe this should be discussed as the primary mechanism for why this intermittent therapy is actually effective in this case, as I believe this is one of the main points of the paper.

We agree fully with your assessment and thank you for pointing out that this was not clear in the previous manuscript version.

- To your point on explanation, we now discuss the results from the model selection in more words (using a more verbal explanation, in line with what the reviewer wrote) and write (line 178): “However, the conclusion that phenotypic adaptation is directed provides a growth rate data-driven explanation for why intermittent treatments here outperform continuous treatments at 500 nM encorafenib over four weeks: the intermittent treatment allows the cells to traverse phenotype

space to drug-sensitive states during drug-off periods, and back to drug-resistant states during drug-on periods. Meanwhile, the continuous treatment traps the cells in highly drug-resistant states. Thus, for the cells, the dynamic adaptation induced by intermittent treatment confers a long-term proliferation disadvantage compared to continuous treatment in the regarded BRAF-mutated melanoma system, with cells pre-incubated at 500 nM encorafenib. This follows from the empirical observation that net growth rates are higher for drug-naive cells when the drug is off, but higher for drug-adapted cells when the drug is on.”

- Moreover, we also consider a simple two-state ($x \in \{0, 1\}$) deterministic model that encodes precisely this binary switch behavior in the supplementary material as a special case of our presented IBM (with the number of intermittent states $n = 0$). Indeed, our IBM and its accompanying ODE analysis (Eqs 4-7) allows for the evaluation of different phenotype update strategies (rather than purely “intelligent” adaptation).

Point 2. I think it would be useful to have a table somewhere (maybe in the SI) which includes all of the parameters used in every simulation/figure. I know there is a GitHub repository link, but I don't think a general reader will want to download the code to actually find parameter values. It would be better to either provide a common table somewhere, and then state this in each figure legend, especially where parameters may deviate from the table. Also, sometimes parameters are assigned inline to a paragraph, which makes them challenging to find if one wants to produce a simulation. A section dedicated to parameters would make this much more clear, in my opinion.

We have updated the Supplementary Material to include:

- A new section (S1): *Model parameters used in main the article results figures*;
- A new table (S1) tabulating the model parameters used to generate all manuscript figures.

These are referred to in the main manuscript (**line 569**): “*The parameter values used to generate the simulation results presented in this article are tabulated in the Supplementary Material (SM1).*”

Point 3. I am a little confused at the top row of Figure 2 (c). What is “day-28” meant to indicate? I also generally do not like the resolution of this image. The trait-space (x-axis) is hard to read, and they are too close together (looks like $x = 0$, $x = 10 - 10 - 1$). I think too much information is presented too compactly, and I think it should be spread out a bit, or maybe overlaid, so as to reduce the number of distinct panels (72).

To make Figure 2c easier to parse we have:

- (a) Moved Figure 2c to a new figure (Figure 4) in the Methods section.
- (b) Made the size of Figure 4 (previously 2c) bigger so that the axes labels are easier to read.
- (c) Made the caption of Figure 4 (previously 2c) more descriptive (see caption below).
- (d) Moved the main text describing Figure 4 (previously 2c) from the Results-section to the Methods-section (**lines 456-465**).

The Figure 4 caption now reads (and explains “-28”): “*The four candidate update strategies result in different dynamic phenotype density distributions. The histograms show how cell population phenotype density distributions change over time in response to no, continuous and intermittent 300 nM BRAF-inhibitor treatments for the four update strategies described in Eqs. 2a-d. Densities are shown with 100 simulations layered over each other (shaded regions) and means (solid bars). 28 days before the experiments start (at day -28), cells are seeded with uniformly distributed phenotype states.*”

As a result of steps (a) and (d) above, we re-ordered two of the Methods-subsections:

- Previously 5th section now 3rd section is: *Modelling cell-level phenotypic adaptation with different update strategies*;

- Previously 3rd section now 5th section is: *Using growth rates to parameterise phenotype and drug dose-dependent fitnesses.*

Point 4. Related to the previous: I don't really understand Figure 2 (e). What does the color mean? And what is the circular shaped supposed to represent? It says in the caption that it represents the "number of death events per cell," but how can a cell have multiple death events? It is not clear to me what information is attempting to be conveyed in this image.

- Following the response to Point 3 above, we have simplified Figure 2 and updated the caption accordingly.
- The "circular shapes" in your feedback refers to illustration of cells, which is now described in the (d)-caption which reads: "(d) The highlighted result in panel (c), together with panel (a), suggest that phenotypic adaptation is directed towards states of high fitness, where this direction depends on the applied drug dose. The schematic shows how cells can adapt from more being drug-sensitive phenotypes (blue) to more drug-resistant phenotypes (red). Drug doses included in the top and bottom arrows respectively induce phenotypic adaptation in the direction of increasing and decreasing x -values."
- We have renamed "number of death events per cells" to "normalised death counts" in Figure 2e.
- Adding to the above point, the meaning of normalised death events is now addressed in the Figure 2 (e)-caption which reads: "(e) The plots show the number of death events normalised over the number of cells over time for the simulation experiments in (c) with directed update strategies in response to 500 nM encorafenib."

Point 5. In Figure 3 (f), the delay in treatment commencement is investigated. When the delay is (a) 1 day (left): $\eta = 1$ is better (1 update/day produces smaller counts) (b) 4 days (middle): $\eta = 4$ is better (4 updates/day produces smaller counts) (c) 10 days (right): $\eta = 1$ is better (1 update/day produces smaller counts) Is there an explanation for this? What is going on that produces this non-monotonic behavior? Should I interpret this in some way, biologically? Is this clinically actionable?

We now describe and explain the non-monotonic behaviour (and its biological implication) in more detail, directly addressing your questions on line 247: *For instance, we see that if the treatment interval length is $T = 1$ day, and the first drug-on period starts after day 1 or 10, then cells with $\eta = 1$ are more treatment-sensitive than cells with $\eta = 4$. However, the opposite is true when the drug-on period starts after day 4. These quantitative specifics follow from the chosen data and model; in particular, whether cells have (or do not have) time to traverse phenotype space in a way that makes them sensitive to sudden changes in drug exposure and removal strongly influences which T and treatment-delay values best suppresses cell proliferation from a human perspective, or equivalently, which η value best promotes survival for the cells. Although the quantitative results presented in Figure 3 are problem-specific, the qualitative demonstration that treatment scheduling together with phenotypic adaptation shape optimal strategies has broader relevance beyond this study and motivates treatment designs that includes monitoring if and how phenotypic expression changes during treatment.*

Point 6. What exactly does the grey-black shading represent in the heat maps in Figure 3? I am not quite sure of the upper bounds used in the coloring, since it looks like white represents 0.0741 as an effective growth rate. I believe the discussion in this section could use a little more detail, as it was hard for me to understand exactly what was being represented, and what the main point of the green box and curve were.

We have substantially revised the third result section to better motivate and explain the study objective and Figure 3 results. Detailed points below.

- We more clearly motivate the objective of the simulation experiment on **line 211**: *‘By comparing the λ_{eff} -values resulting from continuous and intermittent treatments, we can evaluate which treatment is more effective at keeping λ_{eff} low, and thus impeding cell population growth. Using both the simulation and the ODE approach with instantaneous drug on/off switches, we identify (η, T) pairings for which intermittent treatments with 1:1 scheduling such that $t_{\text{on}}=t_{\text{off}}=T$ yield lower effective growth rates than their continuous counterparts. For these (η, T) pairings only, simulated intermittent treatments thus outperform continuous ones long-term, despite resulting in approximately half of the accumulated applied drug.’*

- We more clearly describe the heatmap in the main text (**line 220**): *‘Demonstrating this in Figure 3b,e, (η, T) pairings for which intermittent treatments outperform continuous ones are highlighted in yellow-coloured heatmap regions, whereas such pairings for which the opposite is true are highlighted in black-coloured regions. In the heatmaps, the two regions are separated by overlaid ODE-derived curves that show when the two treatment strategies yield the same effective growth rate. Note that in the 300 nM heatmaps, all visualised (η, T) pairings outperform the continuous treatment strategy, hence no separating curve is shown.’*

- We also describe the heatmap colours more clearly in the Figure 3 caption: *‘In the yellow-coloured bins, intermittent treatments produce long-term effective growth rates lower than those from continuous treatments, and in the black-coloured bins the opposite is true. In the white bins, intermittent and continuous treatments yield approximately equal effective growth rates. From the ODE model approximating these simulations, analytically derived (η, T) pairings where this equivalence holds are shown as neon-dashed curves overlaid on the heatmaps; along these curves, the effective growth rate corresponds to the value indicated by green-dashed frames on the colour bar. Note that the colour scales differ between heatmaps to clearly illustrate the impact of η and T for a fixed dose and initial condition. Note further that in the 300 nM heatmap, all shown (η, T) pairings yield lower effective growth rates than the continuous treatment and thus no neon-dashed curves are seen.’*

Point 7. Minor point: you may want to write (η, T) pairings, as opposed to $\eta - T$ pairing, as the latter looks like subtraction at first pass.

We now write (η, T) pairings.

See highlighted text in Results (lines 198, 200) and the Figure 3 caption.

Point 8. What is λ_* ? I think you should provide a few more details on exactly where equation (1) comes from, and why it is needed, as opposed to just simple exponential growth. That is, what exactly is $\text{FC}_{\text{rel}72}$? You should precisely define it from the data, to make it clear why λ_* is needed. In the same way, I think it would be useful to provide the cost function used to fit the data in Figure 3 (f). I know you mentioned it is least-squares, but writing it precisely (again, in the supplement) would be useful.

- **Response regarding λ_* .**

λ_* is now relabeled λ_{500} for clarity. It is estimated to be 0.1059 day^{-1} . To clarify this, and why we use Eq. 1, we have reworded the corresponding part in the Methods section (**line 381**):

‘The growth rate data used to map out the fitness matrix (Figure 1) are extrapolated from cell count assays performed by Kavran et al.¹ In the experiments, cells are incubated with 500 nM encorafenib prior to a drug challenge with an encorafenib dose of 0, 0.1, 0.3, 1, 3, 10, 30, 100, 300, 1000, 3000, or 10000 nM. The data from the assays are reported in relative fold changes in cell counts. As such, for each drug concentration d , we have data for $\text{FC}_{72}^{\text{rel},d}$ which describes ‘the fold change in cell counts after 72 hours in response to dose d ’ relative to ‘the fold change

in cell counts after 72 hours in response to dose 500 nM'. We use this data to extract λ_d , the effective growth rate for dose d (in nM) between hours 0 and 72, through Eq. 1,

$$\lambda_d = \frac{\log(\text{FC}_{72}^{\text{rel},d})}{3} + \lambda_{500}, \quad (1)$$

where we have assumed exponential population growth over the 72 hour period. The term $\lambda_{500} \approx 0.1059 \text{ day}^{-1}$ is estimated using reported cell count data following a 7 day period of 500 nM treatment, applying linear regression on the log-scale under the assumption that growth is approximately exponential.

• **Response regarding the cost function related to Figure 2 (we assume you mean Figure 2, not 3).**

To clarify our work we now write on (line 500):

“Thus, for each update rule ℓ , the four selected parameter values

$$C_\ell^* = \left(FM(0, 500 \text{ nM}), FM(1, 500 \text{ nM}), \eta_{on}, \eta_{off} \right)$$

that simultaneously minimise the squared distance between the mean model predictions $M_{s,t}$ and in vitro data $D_{s,t}$ are obtained through

$$C_\ell^* = \arg \min_{C_\ell} \sum_{s \in \{\text{cont}, \text{int}\}} \sum_{t=1}^4 \left(M_{s,t}(C_\ell) - D_{s,t} \right)^2,$$

implemented in MATLAB. In Eq. 2, the model-to-data distances are evaluated for the settings continuous and intermittent 500 nM treatments ($s \in \{\text{cont}, \text{int}\}$) at four time points t (excluding the initial condition).”

Point 9. In “Implementing cell division, cell death, and phenotype updates,” you mention that you probabilistically decide when to update cell death/division events (0.5 probability before, 0.5 probability after). What is the point of this? Why didn’t you just pick one rule, and use that throughout? Does this choice have an effect?

The order of operations (whether phenotype-updates or birth-death-events happen first) does not affect our conclusions. Nor does it affect our numerical results in the limit where the cells can take infinitely many phenotype states. However, the order of operations does affect our numerical results when the cells can take several (but not a large number) of phenotype states. These numerical effects impact the comparison with our analytical results.

With the stochastic order-of-operations component (the dice in Figure 2b), we are thus reducing the impact of our priority-choice by alternating between phenotype-update-first and birth-death-event-first.

We now clarify this in the **Figure 2 caption** where we write: “Probabilistic path branching is implemented to minimise numerical bias arising from the ordering of birth–death and phenotype-update events.”

We can illustrate the impact of event-ordering with a simple example:

Consider modelled cells that can be in one of three phenotype states: 0, 0.5 and 1. Let’s say that the cells start in state 0, and deterministically move towards increasing phenotype states at every time point.

Case 1: If phenotype-updates occur before birth-death-events then,

- At time point 1: cells start in state 0, move to state 0.5, divide/die at rate $r(0.5)$.
- At time point 2: cells start in state 0.5, move to state 1, divide/die at rate $r(1)$.

Case 2: If before birth-death-events occur before phenotype-updates then,

- At time point 1: cells start in state 0, divide/die at rate $r(0)$, move to state 0.5.
- At time point 2: cells start in state 0.5, divide/die at rate $r(0.5)$, move to state 1.

Since Cases 1 and 2 generally will give different numerical results, we are reducing the impact of our order-of-operations choice by stochastically alternating between Cases 1 and 2.

Point 10. You define ρ as the probability that a cell changes its phenotype at a programmed update time. But in Methods (top of page 14), you say that the biased update rule can be considered a special case of the semi-biased update rule with $\rho/2 = 1$. This means $\rho = 2$, which is not a probability. You may want to change the wording a bit here.

Thank you for pointing out that mistake. We have corrected this and now write (line 451):

“Note that the no update rule (PU1) can be considered a special case of both the unbiased (PU2) and semi-biased (PU3) update rule with $\rho = 0$. Furthermore, the biased update rule (PU4) can be regarded as the deterministic counterpart of the semi-biased update rule (PU3), in which the probabilistic pre-factor $\rho/2$ is omitted. That is, at each time step, PU3 permits fitness-improving transitions only with probability $\rho/2$, whereas PU4 enforces such transitions with probability one.”

Point 11. It seems like you mostly chose $\rho = 1$ in your simulations. Did you test the effect of different ρ values on your results at all? How robust are your conclusions with respect to ρ ?

In response to your question we now include 3 new figures in the Supplementary Material (Figures S4–6) where we show that our conclusion –that directed phenotype update models better capture dynamic cell count data than the undirected models– is, indeed, robust to a range of investigated ρ -values for $n > 0$. These figures extend the previous purpose of Figure S3, in which we keep $\rho = 1$ but vary n .

To communicate the above result, the following paragraph has been added to the manuscript (line 519): *“For all simulations in the main part of this article, ρ is fixed at $\rho = 1$ and thus the values of η_{on} and η_{off} determine the speed of phenotype adaptation, given a specified n . In the Supplementary Material (SM2), we have re-fitted C_i^* for variations of ρ . As expected, these supplementary results demonstrate a non-linear trade-off relation between ρ , n , and $\eta_{on,off}$: if the probability to accept update propositions (i.e., ρ) decreases, then similar macroscopic system behavior can be recovered by altering the number of times that the cells can update their phenotype states per day, or the number of phenotype states n between the extrema $x = 0$ and $x = 1$. The observation that the directed phenotype update models better capture dynamic cell count data than the undirected models for $n > 0$ is robust to the investigated variations of $\rho \in \{0.25, 0.50, 0.75, 1\}$ and $n \in \{0, 1, 3, 5, 7, 9\}$.”*